# Acute disruption of the synaptic vesicle membrane protein synaptotagmin 1 using knockoff in mouse hippocampal neurons

Jason D Vevea[1,2], Edwin R Chapman[1,2]*

[1]Department of Neuroscience, University of Wisconsin-Madison, Madison, United States; [2]Howard Hughes Medical Institute, Madison, United States

**Abstract** The success of comparative cell biology for determining protein function relies on quality disruption techniques. Long-lived proteins, in postmitotic cells, are particularly difficult to eliminate. Moreover, cellular processes are notoriously adaptive; for example, neuronal synapses exhibit a high degree of plasticity. Ideally, protein disruption techniques should be both rapid and complete. Here, we describe knockoff, a generalizable method for the druggable control of membrane protein stability. We developed knockoff for neuronal use but show it also works in other cell types. Applying knockoff to synaptotagmin 1 (SYT1) results in acute disruption of this protein, resulting in loss of synchronous neurotransmitter release with a concomitant increase in the spontaneous release rate, measured optically. Thus, SYT1 is not only the proximal $Ca^{2+}$ sensor for fast neurotransmitter release but also serves to clamp spontaneous release. Additionally, knockoff can be applied to protein domains as we show for another synaptic vesicle protein, synaptophysin 1.

## Introduction

Protein function is frequently determined through removal or disruption of the protein of interest, followed by analysis of the resultant phenotype. Removal or disruption of proteins can be achieved through a genetic knockout approach (KO) (*Shalem et al., 2014*) or by inhibiting translation of mRNA to knockdown (KD) the protein of interest (*Fire et al., 1998*). Disruption of proteins by KO or KD is not rapid (weeks to days), and long-lived proteins may persist for weeks in post mitotic cells (*Dörrbaum et al., 2018*; *Cohen et al., 2013*). Moreover, genetic and functional compensation will always be a concern during chronic loss of protein function (*El-Brolosy and Stainier, 2017*). Other methods, that rely on protein tags, promise faster protein inactivation (usually in a KO or KD background) (*Natsume and Kanemaki, 2017*). Some of these methods include: chromophore/fluorophore assisted light inactivation (CALI/FALI) (*Lin et al., 2013*; *Marek and Davis, 2002*), auxin-inducible degron (AID) (*Nishimura et al., 2009*), and Trim-Away (*Clift et al., 2017*). Yet another approach, knocksideways, sequesters the protein of interest away from is site of action (*Robinson et al., 2010*). All these methods have benefits and drawbacks. CALI and FALI promise high temporal and spatial control, but the generated inactivating free radical radius is relatively large, so this approach likely affects neighboring molecules. Knocksideways is only applicable to cytosolic proteins, and similarly, AID has only been shown to work with cytoplasmic, nucleoplasmic, and peripheral membrane proteins (*Nishimura et al., 2009*). Trim-Away is a potentially powerful method because it does not require a KO/KD background. However, this approach requires highly specific antibodies and involves harsh electroporation of target cells. Additionally, because mass ubiquitination of membrane proteins is also a strong signal for selective organelle autophagy (*Anding and Baehrecke, 2017*), we developed an alternative approach. There remains an unmet need for a method to conduct non-invasive, rapid, and specific disruption of integral membrane

*For correspondence:
chapman@wisc.edu

**Competing interests:** The authors declare that no competing interests exist.

proteins. Here we describe such a method, which we term knockoff, that uses the Hepatitis C virus (HCV) NS3/4A protease to remove membrane proteins from the bilayer that they are anchored to, leading to acute degradation.

Use of NS3/4A as a tool has been limited to controlling nonreversible events such as split GFP complementation in TimeSTAMP (*Lin et al., 2008*), gene regulation via SMASh tag (*Chung et al., 2015*), protein dimerization with ligand-inducible connection (LInC) (*Tague et al., 2018*), and stabilizable polypeptide linkages (StaPLs) (*Jacobs et al., 2018*). Here, we optimized NS3/4A for druggable cleavage of membrane proteins. Knockoff uses the HCV NS3/4A protease and a small 10 amino acid cleavage site for disruption. If the membrane protein is cleaved on the cytosolic side of the bilayer and exposes an N-degron, the cleaved protein is rapidly degraded by the N-end rule (*Varshavsky, 2017*). We found that previously used combinations of NS3/4A and inhibitors presented issues; they resulted in incomplete cleavage of the target protein in the absence of inhibitor, and incomplete protection in the presence of inhibitor. Our goal for knockoff was to develop a robust system where tightly controlled cleavage results in the acute inactivation of a membrane protein. In this study, we screened commercially available NS3/4A inhibitors for toxicity and fast dissociation from the protease, modified the substrate cleavage site to create a completely on or off system, optimized the codon sequence of NS3/4A for mammalian expression, and demonstrated efficient cleavage (substrate-protease interaction) spanning large protein domains.

We demonstrate the functionality and utility of knockoff by applying the system to synaptotagmin 1 (SYT1). SYT1 functions as a $Ca^{2+}$ sensor for the fast, synchronous release of neurotransmitter in central neurons (*Chapman, 2008*). It is well established that chronic loss of SYT1 in KO neurons results in the loss of synchronous release (*Geppert et al., 1994*; *Littleton et al., 1993*). Moreover, early studies, focused on the neuromuscular junction of *Drosophila* larvae, concluded that loss of SYT1 also resulted in increased rates of spontaneous release (*DiAntonio and Schwarz, 1994*; *Littleton et al., 1993*). This result was the first indication that SYT1 might have a dual function: to clamp or suppress spontaneous release under resting conditions, and then to trigger release in response to $Ca^{2+}$ influx during evoked synaptic transmission. However, subsequent studies, using *Drosophila* embryos, concluded that there was no change in mini frequency, suggesting the mini phenotype in larvae was due to homeostatic mechanisms that come into play during development (*Yoshihara and Littleton, 2002*). Indeed, inhibiting action potential firing of mature neurons leads to increased physical synaptic size (*Murthy et al., 2001*) and increased spontaneous release frequency (*Burrone et al., 2002*). Chronic loss of synchronous neurotransmitter release in *Syt1* KO (S1KO) neurons may result in a similar homeostatic response and explain the increased rate of spontaneous release (mini events) seen in most studies of S1KO neurons. This view became more complicated when SYT1 mouse KO neurons were examined. In the first mouse KO study, changes in spontaneous fusion rates were not observed (*Geppert et al., 1994*), but this appears to be due to the use of autaptic cultures and mini frequencies were in fact elevated in mass dissociated neuronal cultures (*Liu et al., 2009*; *Wierda and Sørensen, 2014*) and in brain slices (*Kerr et al., 2008*). In contrast to the chronic KO approach, acute disruption of SYT1, using CALI/FALI, had little effect on mini frequency (*Marek and Davis, 2002*). This result suggested that increases in spontaneous release rates were the result of compensatory mechanisms using a KO approach. However, mutations that uncouple the function of SYT1 in evoked versus spontaneous release led to the opposite conclusion, and indicated that SYT1 may indeed be a fusion clamp (*Liu et al., 2014*). At present, the idea that SYT1 inhibits spontaneous release, that is the SYT1 clamping hypothesis, is still the subject of vigorous debate.

Here, we develop knockoff and apply it, for the first time, to study the function of SYT1. To monitor both the spontaneous and evoked modes of neurotransmitter release, we used the fluorescent glutamate sensor iGluSnFR (*Marvin et al., 2013*). We observed that acute disruption of SYT1 results in not only the expected loss of synchronous evoked release, but also results in a concomitant rise in spontaneous release rates. These findings validate knockoff and support a role for SYT1 in directly clamping spontaneous release.

## Results

### Synaptotagmin 1 is a difficult to disrupt, long-lived protein

In our first attempt to acutely disrupt SYT1, we expressed CRE recombinase via lentiviral transduction into mature (13DIV) floxed *Syt1* mouse neurons. Expression of CRE causes excision of exon five from this floxed line with loss of *Syt1* transcript, and thus protein. We confirmed CRE transduction at 1DIV resulted in complete loss of SYT1 protein in mature neurons (*Figure 1a*). Mature neurons are more resistant to transduction than immature neurons and for this reason we used a higher titer of lentivirus (10x). However, regardless of the amount of CRE lentivirus used, transduction at 13DIV resulted in incomplete loss of SYT1 protein (*Figure 1a–b*) even though immunostaining of MAP2 and CRE confirmed complete neuronal coverage (*Figure 1c–d*). Strikingly, approximately half of the

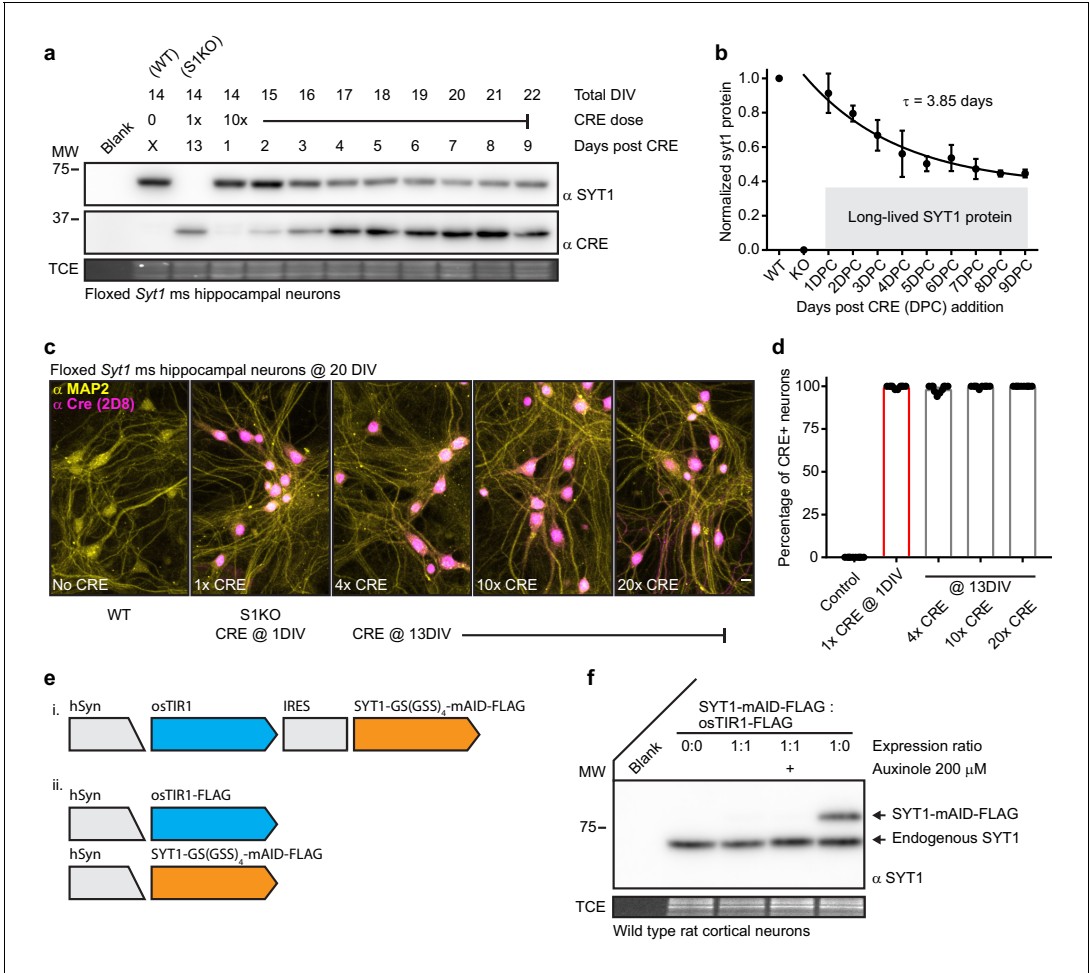

**Figure 1.** Synaptotagmin 1 is a difficult to disrupt, long-lived protein. (a) Representative anti-SYT1 and anti-CRE immunoblot of WT, *Syt1* KO (generated using 1x CRE lentivirus at day 1), and neurons transduced with 10x CRE lentivirus at 13 DIV. Total DIV, CRE dose, and incubation time post CRE transduction are labeled at the top of the blot. Trichloroethanol (TCE) in-gel fluorescence served as a loading control for each immunoblot in this, and all subsequent, figures. (b) Cleavage was quantified via densitometry of the SYT1 signals in panel (a), data were fitted with a single exponential function ($R^2$ = 0.7583), yielding a τ for SYT1 turnover of 3.85 days. The plateau represents a significant population of long-lived SYT1 that was not turned-over. Mean +/- SEM from three trials. (c) Representative confocal fluorescent ICC images from mouse hippocampal neurons at 20 DIV. Images of WT, S1KO, and neurons transduced with 4x, 10x, and 20x CRE lentivirus at 13 DIV, stained with anti-MAP2 (yellow) and anti-CRE (magenta) antibodies. (d) Percentage of CRE positive MAP2 positive soma from indicated conditions (n = 10 for each condition from three independent trials). (e) Schematics of auxin-induced degron expression vectors. (f) Representative anti-SYT1 immunoblot from cultures transduced with vectors shown in e ii; leak prevented detectable expression of the fusion protein, even in the presence of the osTIR inhibitor, auxinole.

The online version of this article includes the following figure supplement(s) for figure 1:

**Figure supplement 1.** Characterizing expression levels of AID components in neurons.

SYT1 protein appeared to be lost in 4 to 5 days but the other fraction showed no detectable turnover during our analysis period, greater than 1 week (*Figure 1b*). Therefore, SYT1 is a long-lived synaptic protein with a substantial population of molecules that are resistant to turnover. The discovery of two pools of SYT1 that have very different half-lives highlights the need to target the protein, itself, for degradation. So, we attempted to degrade SYT1 directly using the established auxin-inducible degron (AID) technique (*Natsume et al., 2016*). We first constructed a lentiviral IRES expression vector based on a recently published construct (*Zotova et al., 2019*). This construct was modified to express mAID-tagged SYT1 along with the E3 ubiquitin ligase osTIR1 (*Figure 1e* i.). However, we could not detect mAID-tagged SYT1 (data not shown). Therefore, we split osTIR1 and the mAID-tagged SYT1 into separate vectors in order to better control expression levels of each (*Figure 1e* ii. and *Figure 1—figure supplement 1a–b*). We found that the mAID-tagged SYT1 was not stable in the presence of osTIR1, indicating leak. Even addition of the osTIR1 inhibitor, auxinole, could not stabilize mAID-tagged SYT1 (*Figure 1f*). Given these observations, we can only reason that there is leak in the AID system and that this leak becomes an issue when studying long lived proteins in post-mitotic cells. This may be why AID technology has not been applied to neuronal targets. Indeed, leak in the AID system was recently documented (*Yesbolatova et al., 2019*); and moreover, the entire family of TIR1/AFB proteins all seem to have a significant auxin independent interaction with their targets, limiting the utility of this method (*Parry et al., 2009*). Thus, there is a compelling need for the creation of a well performing technology to control protein levels, especially long-lived membrane proteins.

## Knockoff development using a model substrate

For development of knockoff, a model self-cleaving protein was used, as illustrated in *Figure 2a*. The model self-cleaving protein (SELF-mito) was targeted to the mitochondrial outer membrane (OMM) by the C-terminal targeting sequence from OMP25 (*Nemoto, 1999*). This was followed by: a cleavage site, green fluorescent protein with monomeric and superfolder mutations (msGFP) (*Pédelacq et al., 2006*; *Zacharias et al., 2002*), and the NS3/4A protease (*Shimizu et al., 1996*). The P6P4' NS5A/5B cleavage sequence was selected because it is both the smallest and fastest NS3/4A substrate (*Zhang et al., 1997*). The inhibitor used for knockoff should be completely non-toxic and dissociate quickly. Inhibitors that were covalent or showed toxicity in HEK293T cells were not investigated further (*Supplementary file 1*). Only Danoprevir (DNV), Paritaprevir (PRV), and Glecaprevir (GCV) were non-toxic at all doses, indicating a high-safety margin (*Figure 2—figure supplement 1a*). At a low, 5 μM dose, neuronal apoptosis was not observed (*Figure 2—figure supplement 1b*).

Because the knockoff model protein is anchored to mitochondria via the carboxy terminus, once cleavage occurs, the protein becomes cytosolic and is not degraded by the N-end rule. In this case, the stability of the cleavage product makes it possible to summate breakthrough cleavage (i.e. leak) over long time scales and assay system leak. Using the NS5A/5B cleavage site (EDVVCC/SMSY), commercially available inhibitors were screened for cleavage protection. Only GZV (Grazoprevir) and GCV inhibited cleavage, and this effect was incomplete (*Figure 2b*). The higher immunoreactive band represents mitochondrial bound, uncleaved protein, while the lower band comprises the cytosolic, cleaved protein. We reasoned that modification of the NS5A/5B cleavage sequence might decrease breakthrough cleavage. Using published reports of cleavage site analysis as a guide (*Shiryaev et al., 2012*; *Zhang et al., 1997*), various modified cleavage sequences were tested (data not shown). Substitution of the P1' serine with a glutamine residue (EDVVCC/QMSY) decreased leak and resulted in more efficient protection from cleavage with all of the inhibitors that were tested (*Figure 2b*). Moreover, glutamine is a strong signal for N-end mediated degradation from yeast to mammalian cells, so the presence of this residue should facilitate protein degradation after cleavage of amino-terminally cleaved substrates. Mitochondrial localization of SELF-mito was confirmed using super-resolution fluorescence microscopy of HEK cells maintained in 5 μM PRV; in the absence of PRV the signal was cytosolic, thus establishing the feasibility of knockoff (*Figure 2c*).

The cleavage rate of our model knockoff protein in neurons was examined following removal of the protease inhibitors DNV, PRV, GZV, and GCV. Only DNV and PRV washout resulted in cleavage products within the first hour (*Figure 2—figure supplement 1c*), suggesting that these inhibitors rapidly dissociate from the NS3/4A protease. Cleavage products accumulated quickest upon PRV washout, with almost complete cleavage within the first hour.

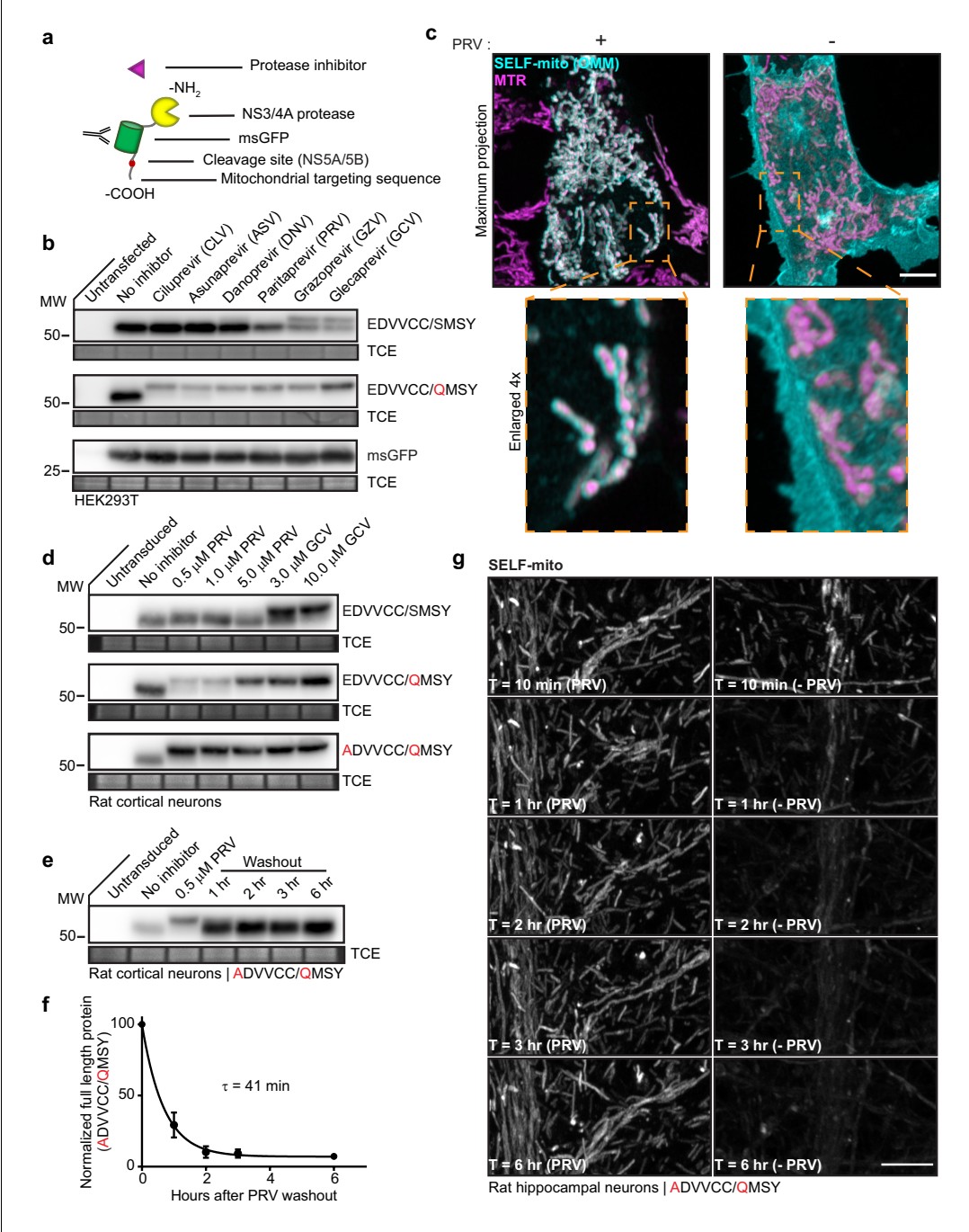

**Figure 2.** Knockoff development using a model substrate. (**a**) Schematic of the model substrate (SELF-mito) used to develop knockoff. (**b**) Representative anti-GFP immunoblots of HEK293T cells expressing model substrates in the presence of the indicated inhibitors. The upper band represents full-length substrate protein; the lower band is the cleavage product. The negative control was untagged msGFP. (**c**) Super-resolution fluorescent images of HEK293T cells expressing SELF-mito with the modified cleavage sequence (EDVVCC/QMSY) counterstained with MitotrackerRed (MTR), +/- 5 μM PRV. Maximum projection and enlarged inset examples are shown. (**d**) Knockoff in neurons required further attenuation of the cleavage site. Representative anti-GFP immunoblots from rat cortical neurons expressing substrates bearing the natural NS5A/5B cleavage sequence (EDVVCC/SMSY), modified sequence (EDVVCC/QMSY), and further attenuated sequence (ADVVCC/QMSY) in the presence of inhibitors. GCV was the most effective inhibitor of substrate cleavage. (**e**) Representative anti-GFP immunoblot from rat cortical neurons expressing the attenuated (ADVVCC/QMSY) model substrate showing the time-course of cleavage after washout of 0.5 μM PRV. (**f**) Self-cleavage time-course of the attenuated (ADVVCC/QMSY) model knockoff protein after inhibitor washout. Cleavage in (**e**) was quantified by densitometry and the data fitted with a single exponential function to yield a τ = 41 min (R² = 0.9675). Mean +/- SEM from three independent trials are shown. (**g**) Representative super-resolution fluorescence images of

*Figure 2 continued on next page*

*Figure 2 continued*

attenuated (ADVVCC/QMSY) substrate (SELF-mito) self-cleaving in a dendrite from a rat hippocampal neuron. (see another example *Video 1*). Scale bars, 5 µm.

The online version of this article includes the following figure supplement(s) for figure 2:

**Figure supplement 1.** Knockoff development using a model substrate.

Next, synaptic function was monitored in the presence of PRV with the intensity based fluorescent glutamate reporter, iGluSnFR (*Marvin et al., 2013*). iGluSnFR is targeted to the plasma membrane where it reports the presence of the neurotransmitter glutamate. For these experiments, iGluSnFR with superfolding mutations (*Marvin et al., 2018*) was subcloned into a lentiviral vector and expressed using a CamKII promoter. ER and Golgi export sequences were also added to the carboxy terminus to improve trafficking to the plasma membrane (*Parmar et al., 2014*; *Stockklausner et al., 2001*). We confirmed that iGluSnFR reliably reported evoked glutamate release when stimulating neurons and varying the extracellular $[Ca^{2+}]$ (*Figure 2—figure supplement 1d*).

PRV is the ideal inhibitor for this system because it dissociates so rapidly. However, 5 µM PRV increased spontaneous network activity (*Figure 2—figure supplement 1e*), so we explored lower doses and found that 0.5 µM drug had no effect on spontaneous activity or synchronous neurotransmitter release (*Figure 2—figure supplement 1f*). Neither high nor low doses of PRV altered synapse number or morphology (*Figure 2—figure supplement 1g*). Because high doses of PRV had effects, we further tuned the NS5A/5B substrate sequence so that low doses (i.e. 0.5 µM) would be effective. This led to the (ADVVCC/QMSY) substrate sequence. This sequence performed extremely well, with full inhibition of cleavage at 0.5 µM PRV (note the effectiveness as compared to 10 µM GCV) (*Figure 2d*). Next, substrate cleavage was monitored via immunoblot analysis to estimate knockoff cleavage kinetics (*Figure 2e–f*). Full-length SELF-mito was quantified using densitometry, plotted, and fitted with a single exponential function, yielding a $\tau$ of 41 min, and a $t_{1/2}$ of 28 min (*Figure 2f*).

Cleavage was also monitored by fluorescence microscopy using fast Airyscan imaging. Representative images, corresponding to the immunoblot time points, were chosen and displayed from an image sequence that acquired Z-stacks every 10 min through rat hippocampal dendritic arbors, for 6 hr in total (*Figure 2g* and *Video 1*). Apparent loss of msGFP signal is from dilution of the construct as it diffuses from the OMM to the cytosol. Loss of the fluorescence was not observed when the inhibitor was included in imaging media (*Figure 2g*).

## Rapid, efficient, targeted degradation of Synaptotagmin 1

SYT1 is an ideal candidate for application of knockoff; it is targeted to synaptic vesicles where it functions as a $Ca^{2+}$ sensor that triggers synchronous neurotransmitter release (*Geppert et al., 1994*; *Nishiki and Augustine, 2004*). As noted in the Introduction, in addition to functioning as a sensor for synchronous neurotransmitter release, there is a debate as to whether SYT1 also serves as a fusion clamp that inhibits the rate of spontaneous neurotransmitter release under resting conditions. Here, we exploit knockoff to determine whether SYT1 acts directly as a fusion clamp that controls spontaneous release.

The general protocol for SYT1 knockoff is illustrated in *Figure 3a*. For S1KO experiments, we relied on *Syt1* floxed mice. The S1-SELF construct has the cleavage site inserted in the juxtamembrane region at amino acid residue 100. The following elements were appended onto the carboxy terminus: a long flexible gly-ser-ala linker, the codon optimized NS3/4A protease, and a single FLAG tag. The S1-SELF construct

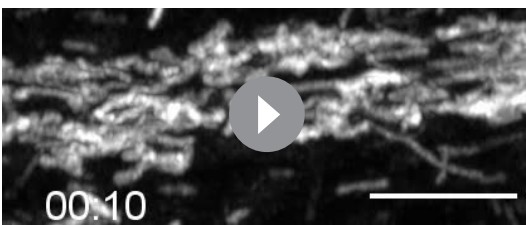

**Video 1.** Knockoff development using a model substrate a, Images of SELF-mito being cleaved in a dendrite from a rat hippocampal neuron. Images acquired using fast Airyscan mode on a Zeiss 880 LSM, with 10 min imaging intervals for six total hours. Video from *Figure 2g*. After cleavage, the model (msGFP) is liberated from the mitochondrial membrane. Scale bar, 5 µm.

https://elifesciences.org/articles/56469#video1

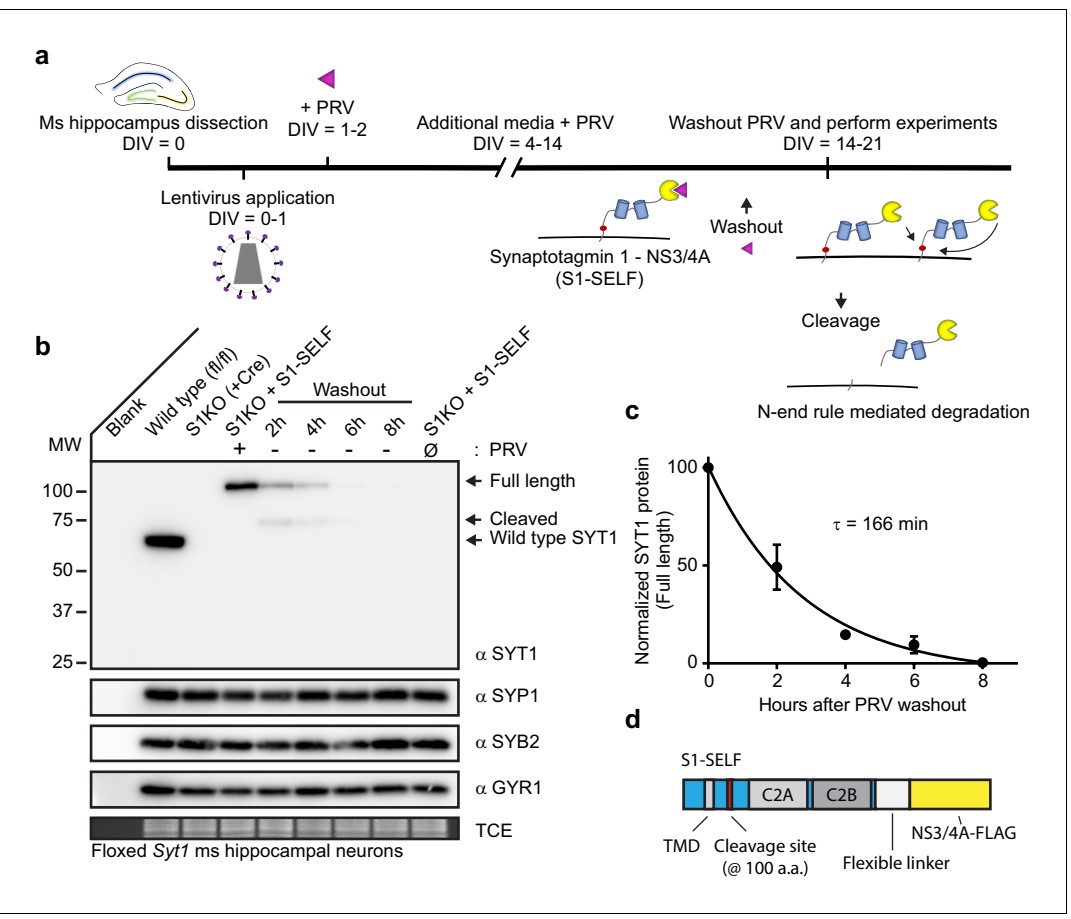

**Figure 3.** Rapid, efficient, targeted degradation of synaptotagmin 1. (**a**) Illustration of the SYT1 knockoff protocol. At 14–21 DIV, inhibitor was removed from neuronal cultures and experiments were performed. Following inhibitor washout, cleavage reactions can occur in cis or *trans*, and the resulting cleavage product is degraded via the N-end rule. (**b**) Representative anti-SYT1, anti-synaptophysin 1 (SYP1), anti-synaptobrevin 2 (SYB2), anti-synaptogyrin 1 (GYR1) immunoblots of WT, SYT1 KO (generated using a CRE virus), and KO neurons expressing S1-SELF, in mouse hippocampal neurons at 14 DIV. (Ø) denotes a condition in which cultures have never been exposed to PRV. (**c**) Self-cleavage time course of S1-SELF, upon PRV washout. Cleavage was quantified via densitometry of the SYT1 immunoblots in panel **b**, and plotted. Mean +/- SEM from three independent trials are shown, and the time constant was determined by fitting the data with a single exponential function ($R^2$ = 0.9498). The τ for S1-SELF cleavage was 2 hr 45 min. (**d**) Schematic of S1-SELF; domains are labeled.

The online version of this article includes the following figure supplement(s) for figure 3:

**Figure supplement 1.** NS3/4A codon optimization and application to the carboxy terminus of synaptophysin.

was introduced using lentivirus, and wild type (WT) levels of expression could be maintained indefinitely when 0.5 µM PRV was included in the media (*Figure 3—figure supplement 1a*). We found it difficult to express high levels of the S1-SELF construct; indeed, the codon usage of NS3/4A was not optimized for mammalian expression (*Figure 3—figure supplement 1b–c*). So, we recoded the NS3/4A protease and observed robust expression of S1-SELF; the recoded construct was used in all subsequent experiments.

Immunoblot analysis of KO neurons expressing S1-SELF revealed rapid, complete, cleavage of SYT1 after washout of PRV (*Figure 3b*). This was specific, as other synaptic vesicle proteins were unaffected (*Figure 3b*). The levels of full-length S1-SELF were quantified using densitometry, plotted, and fitted with a single exponential function, yielding a τ of 166 min ($t_{1/2}$ = 114 min; $R^2$ = 0.9498) (*Figure 3c*). For clarity, a diagram of the S1-SELF construct, is provided (*Figure 3d*).

Knockoff is not just a whole protein disruption method; indeed, well-behaved molecular scissors would find a myriad of uses in cellular biology. We demonstrate just one of these additional uses by

preferentially disrupting a single domain of another SV protein, synaptophysin 1 (SYP1) (*Figure 3—figure supplement 1d*). Application of knockoff to the carboxy tail of SYP1 results in degradation of this domain within 24 hr and provides proof of concept for knockoff application to specific domains of proteins in cells (*Figure 3—figure supplement 1e*).

## Distinct fates of the S1-SELF cleavage products

Taking advantage of antibodies that recognize different epitopes of SYT1, in conjunction with super resolution Airyscan imaging, we addressed the fate of the two cleavage products (*Figure 4a*). Immunocytochemistry (ICC) allowed us to visualize: 1) synaptic vesicle clusters using synaptophysin as a general SV marker, 2) the cytoplasmic domain of SYT1, using an antibody to the C2A domain, and 3) the integral membrane portion of SYT1, using an antibody to the luminal domain (LD) (*Figure 4b*). Note that the LD antibody recognizes rat SYT1 with higher affinity than the mouse protein; hence, the WT sample is only weakly stained (note: S1-SELF was constructed using the rat protein sequence and is recognized by this antibody). These ICC experiments also confirmed that knockoff was not associated with major morphological changes in the SV pools. The redistribution of S1-SELF, as it was cleaved, was quantified using the Pearson's correlation coefficient (PCC). When expressed in KO neurons, both the cytoplasmic domain (C2A) and LD of S1-SELF, in the presence of 0.5 µM PRV, were highly colocalized with synaptophysin. Following PRV washout, the correlation between synaptophysin and SYT1 (C2A) steadily decreased. Interestingly, a small number of SYT1 (C2A) puncta remained visible at later time points, but these did not contain synaptophysin. Conversely, the correlation between synaptophysin and SYT1 (LD) remained relatively high even though a portion of the SYT1 (LD) signal spreads from synaptophysin-positive pools (*Figure 4c*). This observation suggests that the membrane segment of SYT1 is slowly lost to surrounding membranes in the absence of the C2 domains. Importantly, synapse number and size were no different between experimental groups and wild-type neurons (*Figure 4—figure supplement 1a–b*).

## Loss of synaptotagmin 1 via knockoff disrupts synchronous release

After confirming, via immunoblot and ICC, that S1-SELF is rapidly cleaved and degraded following inhibitor washout, we sought to measure changes in synchronous glutamate release. To address this, the fluorescent glutamate reporter iGluSnFR (*Marvin et al., 2018*) was transduced into WT, *Syt1* KO (generated using a CRE virus), and KO neurons expressing S1-SELF. After synapse maturation, neurons were subjected to a single-field stimulus in 1 mM extracellular $Ca^{2+}$, and dendritic arbors expressing iGluSnFR were imaged at 100 Hz. Signals from representative regions of interest (ROIs) from a single field of view (FOV) are plotted in *Figure 5a* (black); averages are shown in green. Loss of synchronous release was apparent from the distribution of iGluSnFR ($\Delta F/F_0$) peaks. To quantify this effect, peak fluorescence changes were quantified and binned (10 ms) against time across the entire data set (*Figure 5b*). Ten ms bins have been used before to quantify synchronous and asynchronous release from WT and S1KO neurons based on electrophysiological measurements (*Yoshihara and Littleton, 2002*). There was no difference in the distribution of iGluSnFR peaks between WT and S1KO + S1-SELF + PRV (S1-SELF protected) following a single stimulus (majority of events are synchronous). In contrast, in S1KO neurons, synchronous release was almost completely absent. These results validate our use of the modified SYT1 construct (S1-SELF). There was no significant difference in the distribution of iGluSnFR peaks between S1KO and the 8 hr washout condition, consistent with complete loss of full length SYT1 protein (statistics summarized in *Figure 5—source data 1*). The findings reported here, using an optical approach, agree with a previous report concerning the fraction of synchronous release in WT and S1KO neurons (*Nishiki and Augustine, 2004*). Average traces and normalized traces are included in *Figure 5—figure supplement 1a–b* and show increased asynchronous release in KO and washout neurons.

Finally, iGluSnFR enabled the localization of glutamate release. Temporally color-coded max projections of the first 100 ms after a single stimulus are shown in *Figure 5c*. Colorized puncta are glutamate release events. Red/orange puncta are early, more synchronous events, while green, blue, and purple correspond to later, asynchronous events. This analysis further documents the time-dependent loss of synchronous release upon cleavage of S1-SELF.

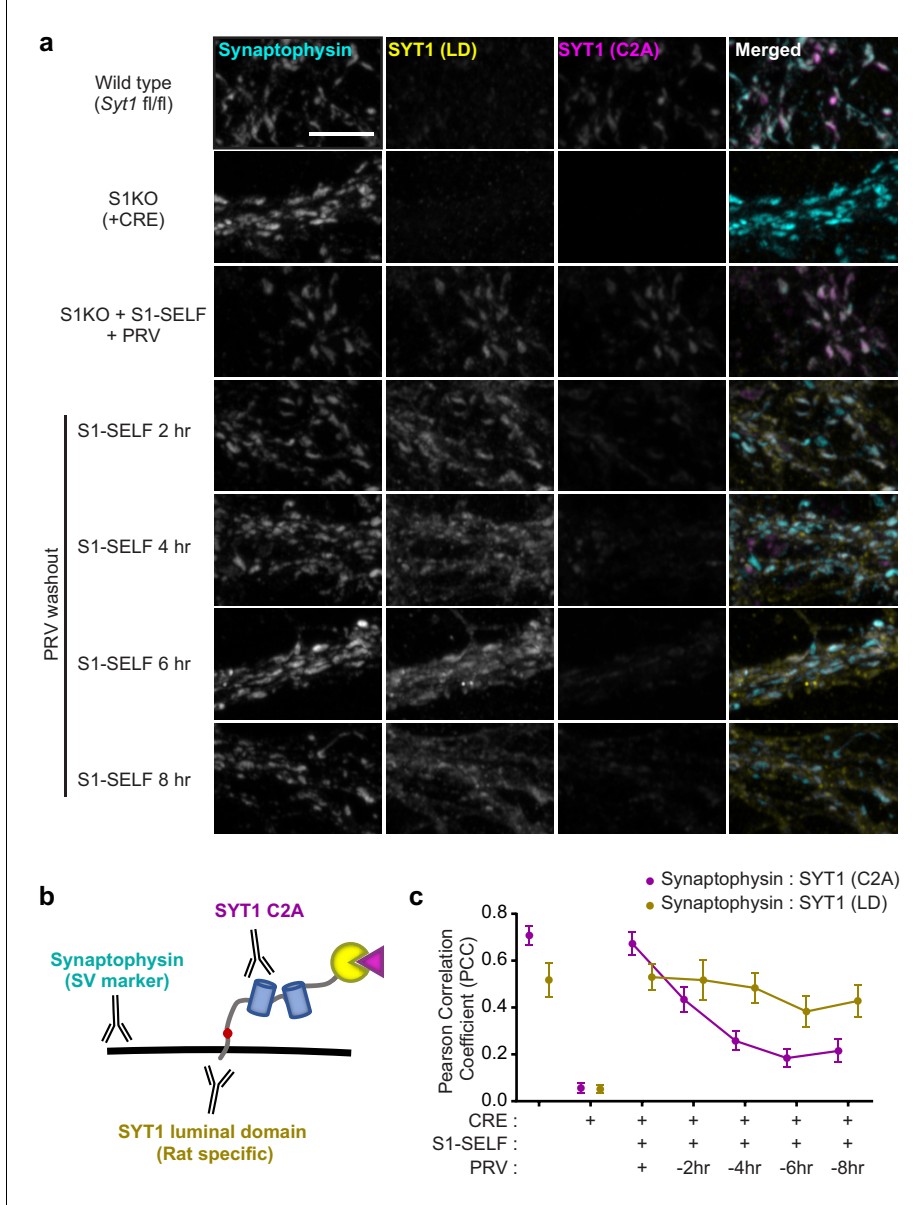

**Figure 4.** Distinct fates of the S1-SELF cleavage products. (a) Representative super-resolution fluorescent ICC images from mouse hippocampal neurons at 15 DIV. Images of WT, S1KO, and neurons expressing S1-SELF, stained with anti-synaptophysin (cyan), anti-SYT1 luminal domain (LD) (yellow), and anti-SYT1 C2A domain (C2A) (magenta) antibodies; in the last column, all three signals were merged. Note the SYT1 LD antibody recognizes the rat luminal domain with much higher affinity than the mouse; therefore, in the WT example, only a trace signal is detected. All images were acquired using the same microscope settings, and all samples were prepared in parallel. Scale bar, 5 μm. (b) Illustration of S1-SELF, and the antibodies used for ICC. (c) Pearson's correlation coefficient (PCC) plot, measured using JaCoP for ImageJ (*Bolte and Cordelières, 2006*). As S1-SELF is cleaved, the PCC of the synaptophysin to SYT1 C2A signal decreases (Mean PCC +/- SEM are plotted, n = 10 for each condition from three independent trials).

The online version of this article includes the following figure supplement(s) for figure 4:

**Figure supplement 1.** Synaptic number and size are not altered during knockoff of SYT1.

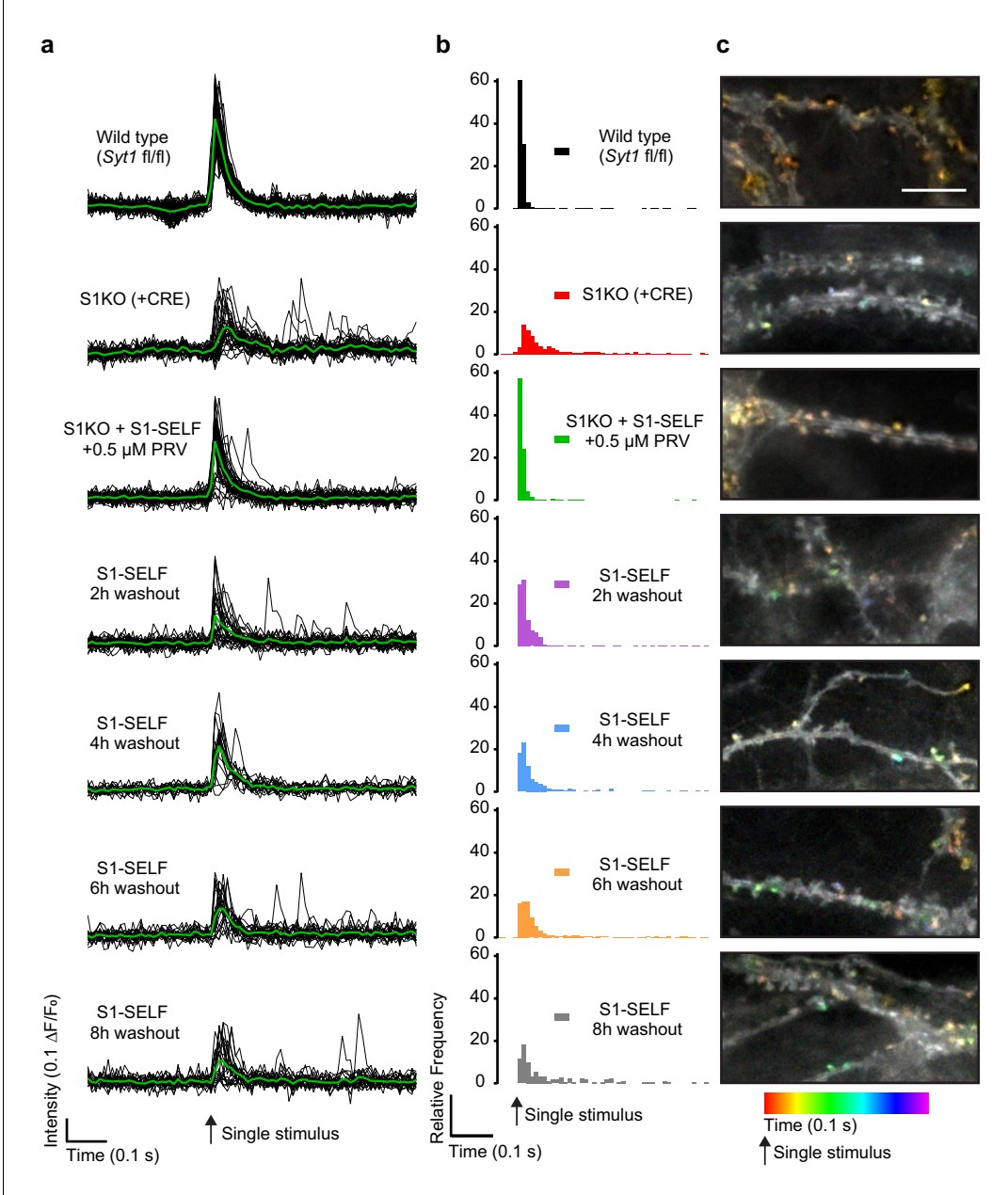

**Figure 5.** Loss of S1-SELF via knockoff disrupts synchronous release. (**a**) Representative, individual ROI (black) and average (green) iGluSnFR traces from a single-field stimulus, recorded optically at 100 Hz. The examples shown are from a single FOV for each condition. (**b**) Histograms of iGluSnFR ($\Delta F/F_0$) peaks plotted using 10 ms bins. Peaks were binned over the entire 1.5 s of recording; a 0.5 s epoch, just before and after the stimulus, is shown. Samples were color-coded as follows: WT (black), *Syt1* KO (+CRE) (red), S1KO + S1-SELF + 0.5 µM PRV (green); the S1-SELF PRV washout samples were: 2 (purple), 4 (blue), 6 (orange), and 8 hr (grey). The histograms include all combined data from four independent trials, with 10 to 16 FOVs for each group. Comparisons between all conditions, and statistical analysis, are provided in *Figure 5—source data 1*. (**c**) Representative images showing temporally color-coded max projections (time projection) of iGluSnFR $\Delta F/F_0$ peaks 100 ms after a single stimulus. Temporal color code is from red (time of stimulation) to purple (100 ms after the stimulus). Scale bar, 5 µm.

The online version of this article includes the following source data and figure supplement(s) for figure 5:

**Source data 1.** Loss of SYT1 via knockoff disrupts synchronous release.

**Figure supplement 1.** Average normalized synchronous glutamate release measured optically during knockoff of SYT1.

## Acute knockoff of SYT1 unclamps spontaneous release

We next examined spontaneous release during inhibitor washout, to address the question of whether SYT1 has a second function as a fusion clamp. We used iGluSnFR to optically measure spontaneous release because: 1) this approach allows for higher throughput of measurements and 2) yields unprecedented spatial information regarding the distribution of spontaneous release sites. However, because iGluSnFR has not been used to measure spontaneous release, we first sought to record AMPA-receptor-mediated miniature postsynaptic currents (mEPSCs) using conventional patch-clamp measurements. mEPSCs were recorded in the presence of 1 µM TTX from WT, S1KO, S1KO + S1-SELF +/- PRV mature mouse neurons. Neurons that had been without PRV for 4 hr showed a clear increase in mEPSC frequency, analogous to the elevated frequency using S1KO neurons (*Figure 6a–b*; statistics are summarized in *Figure 6—source data 1*). The amplitude and kinetics of mEPSCs were similar between all groups (*Figure 6c–d*). We then attempted to record spontaneous glutamate transients using sf iGluSnFR. Optical detection offers not only spatial information but also allows a much higher throughput, allowing us to record from more time points during the knockoff protocol. Neurons were imaged, at various time points, for 1 min in the presence of 1 µM tetrodotoxin (TTX). Temporally color-coded max projections, after background subtraction, are displayed in *Figure 6e*. Spontaneous events near the start of the recordings appear red and orange, and events near the end are pseudo colored blue or purple. Because spontaneous events yielded low signals, they were counted manually. An example trace from an ROI in a S1KO + S1-SELF, 6 hr after washing out PRV, is shown in *Figure 6—figure supplement 1a* where green arrows mark individual events (see also *Video 2*, where the arrow indicates the region used for the trace shown in *Figure 6b*). A larger region from the same sample, showing a high frequency of spontaneous events, is also shown in *Video 3*.

To validate ROI selection, the peak fluorescence change (ΔF) during the time series was plotted for miniature glutamate transient (mGT) events (green) and for ROIs that were offset from release sites (e.g. the background; grey) (*Figure 6—figure supplement 1b*); the histogram peaks were clearly separated. Moreover, plotting the peak amplitude of events versus the signal-to-noise of the same ROI reveals clear separation between mGT events and background ROI peaks (*Figure 6—figure supplement 1c*). The frequency of spontaneous mGT events was 2.4-fold higher in S1KO neurons relative to WT and was unchanged in S1KO neurons expressing S1-SELF in the presence of PRV (protected). During washout, the rate of spontaneous release increased until it plateaued at 4 hr (*Figure 6f*; statistics are summarized in *Figure 6—source data 2*). Similarly, quantifying synchronous release percentage (events within the first 10 ms after single stimulus), shows a similar trend (data from *Figure 5*). Again, most release events in WT, and S1KO neurons expressing S1-SELF in the presence of PRV, were synchronous; S1KOs were largely devoid of these fast responses (*Figure 6g*; statistics are summarized in *Figure 6—source data 3*). To further ensure that homeostatic mechanisms were not invoked, we determined the relative protein levels of a known regulator of homeostatic scaling, CDK5 (*Kim and Ryan, 2010*). Immunoblot analysis revealed no changes in CDK5 expression levels in any of the samples (WT, S1KO, protected, washout conditions) (*Figure 6—figure supplement 1d*). Finally, the percentage of synchronous release was plotted against the spontaneous release rates and a clear relationship emerged. As synchronous release is lost, spontaneous release increased (*Figure 6h*; linear regression yielded an $r^2$ = 0.7688; p-value=0.0097). Because these two variables are correlated, we conclude that SYT1 directly controls both processes.

## Discussion

In the current study, we report a general method to efficiently cleave targeted membrane proteins with high temporal control. Major issues regarding inducible KOs and shRNA-mediated knock down are that they are slow acting, limited by the half-life of the targeted protein, and are usually irreversible. Knockoff overcomes these limitations and provides exquisite specificity. The combination of HCV NS3/4A protease (codon optimized), inhibitor (PRV), and cleavage sequence (ADVVCC/QMSY), were used here to create a cleavable system that is completely inhibited in the presence of inhibitor. Importantly, cleavage was rapid and complete upon inhibitor washout. Other combinations of protease and cleavage sequence performed poorly because either: 1) the protease did not express well, 2) the inhibitor failed to completely prevent cleavage, 3) the inhibitor was toxic or had off-target effects, 4) the protease did not completely cleave all copies of the targeted protein, or 5) the

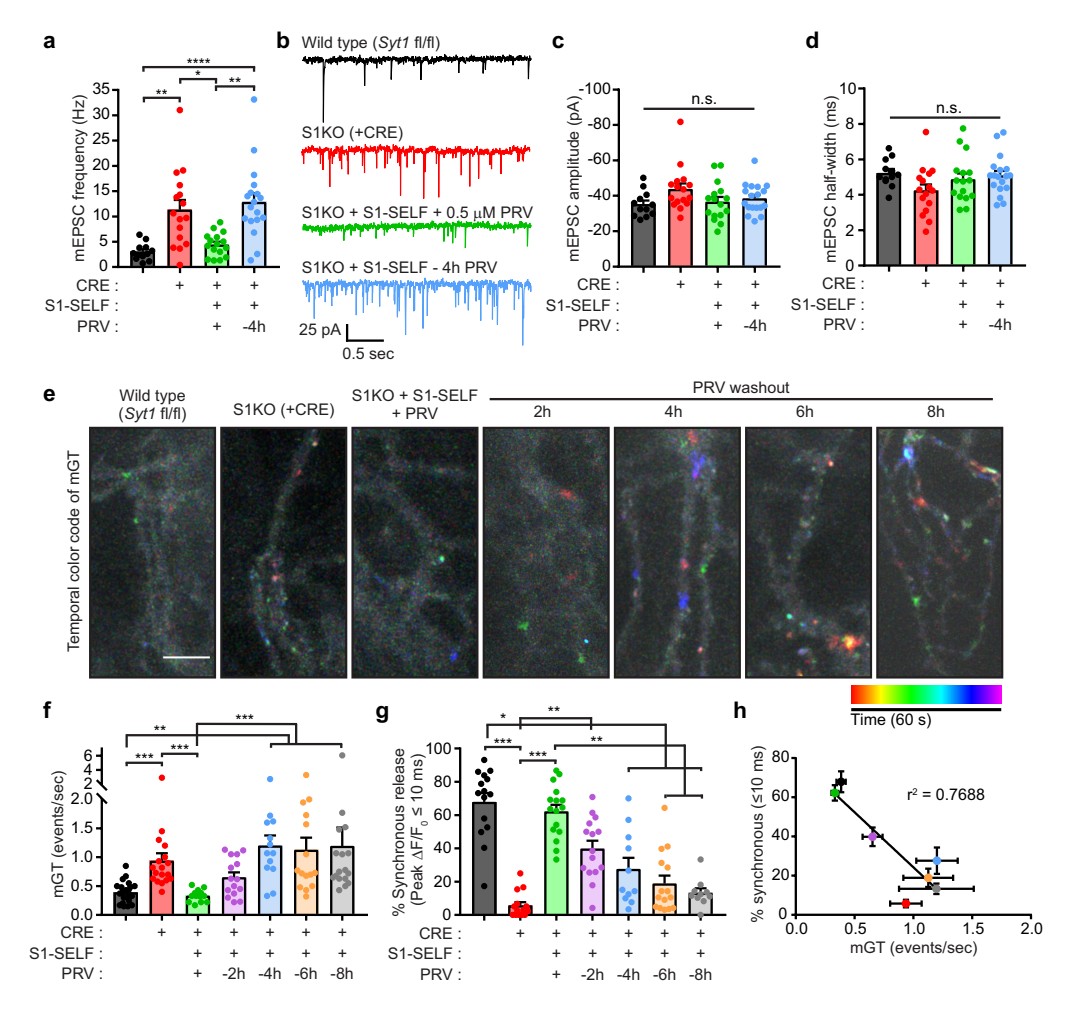

**Figure 6.** Acute knockoff of SYT1 unclamps spontaneous release. (a) mEPSC rates in WT (black), S1KO (red), and neurons expressing S1-SELF (+ PRV green and −4 hr PRV blue). Values are mean +/- SEM, 12 to 18 neurons per condition (n). Comparisons between conditions and statistical analysis, are provided in *Figure 6—source data 1*. (b) Representative traces from (a). (c) Average mEPSC amplitude. (d) Average mEPSC 10–90% half-width. No statistical differences between groups in (c) or (d) were observed. (e) Representative images showing temporally color-coded max projections (time projection) of iGluSnFR ΔF peaks. mGT denotes miniature glutamate release events. Temporal color code is from red (start of image acquisition) to purple (60 s after image acquisition start). Scale bar, 5 μm. (f) mGT rates in WT, S1KO, and neurons expressing S1-SELF, recorded optically with iGluSnFR. Values are mean +/- SEM from three independent trials, with 13 to 21 field of views for each group. Comparisons between conditions and statistical analysis, are provided in *Figure 6—source data 2*. (g) Percentage of iGluSnFR ΔF/F$_0$ peaks within 10 ms following a single stimulus (data are from *Figure 5*). Values are mean +/- SEM from four independent trials, with 10 to 16 field of views for each group. Comparisons between conditions and statistical analysis, are provided in *Figure 6—source data 3*. (h) Inverse correlation between synchronous release (≤10 ms) and spontaneous (mGT) release in WT, S1KO, and S1-SELF neurons. Data were fitted using a linear regression; $r^2 = 0.7671$, p-value=0.0097.

The online version of this article includes the following source data and figure supplement(s) for figure 6:

**Source data 1.** Acute knockoff of SYT1 unclamps spontaneous release as determined electrophysiologically.
**Source data 2.** Acute knockoff of SYT1 unclamps spontaneous release via iGluSnFR.
**Source data 3.** Acute knockoff of SYT1 decreases synchronous release (increases asynchronous release).
**Figure supplement 1.** Identification of spontaneous glutamate transients and probing homeostatic plasticity markers during knockoff.

cleavage reaction proceeded slowly upon inhibitor washout. This is also the first use of the HCV NS3/4A protease in a system where the cleavage site is not immediately adjacent to the protease.

More generally, acute protein disruption has been a major goal in cellular biology. This is a particularly important issue in neuroscience, as neurons are post-mitotic and synapses are highly plastic. Fast disruption of a target protein greatly increases confidence in any observed resultant phenotype. For decades, researchers have sought to develop new tools to achieve this goal, but the results have

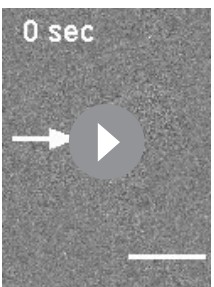

**Video 2.** Acute knockoff of SYT1 unclamps spontaneous release a, Example image sequence of background-subtracted, iGluSnFR ΔF spontaneous events, from S1-SELF 6 hr after washout neuron; four spontaneous events were detected (green arrows). The arrow indicates the position of the ROI. The image sequence and associated data are from the ROI indicated in *Figure 6—figure supplement 1a*. Scale bar, 5 μm.

https://elifesciences.org/articles/56469#video2

been mixed. In the current study, we directly compared our newly developed protocol to acute introduction of CRE in a floxed strain and to an Auxin induced degron (AID) system. We show that, in agreement with other reports (*Dörrbaum et al., 2018*), SYT1, like other presynaptic proteins, is long-lived and hence difficult to disrupt genetically once the protein has been translated. Moreover, we demonstrate significant leak in the AID system, rendering this method unusable in postmitotic cells that have large numbers of stable proteins.

Other groups have developed protease-based systems for acute protein control including SMASh tag, a self-excising degron tag (*Chung et al., 2015*) and TIPI, TEV controlled protein cleavage (*Taxis et al., 2009*). SMASh tag is a an effective tool for turning on or off expression of a protein. However, once the protein is made (in the absence of inhibitor) it cannot be regulated because the degron has been excised and the protein of interest is now governed by its normal cellular half-life. TIPI uses the TEV protease controlled genetically through a selective promoter. Unfortunately, TEV is a 'sloppy' protease that cleaves a number of endogenous targets; moreover, promoters are inherently slow, and many promoters require changing carbon source or have toxic side effects (glucose/galactose and TET, respectively).

The difficulty in creating a robust technique that allows druggable control of a protein, and the creation of acute protein disruption techniques, are major subjects of discussion in the neuroscience community (*Südhof, 2018*). In neuroscience in general, and regarding SYT1 function in particular, relying on KO methods has confounded the true function of proteins in synaptic transmission for decades. Knockoff addresses this need by providing the means to acutely disrupt a protein of interest, at its terminal destination, by: 1) demonstrating that the NS3/4 protease can cut at a cleavage site that is not immediately adjacent to itself (in fact, it can cleave across large protein domains such as the C2 domains of SYT1), 2) screening inhibitors of NS3/4 protease for toxicity and fast dissociation rates, 3) modifying the substrate site so that low doses of a fast dissociating inhibitor completely protect the modified protein (100% on or off), and 4) providing evidence that the modified protein is protected from cleavage for weeks at a time (no leak from our system, unlike other systems). This proof of principle, demonstrating the separation of protease and cleavage site, should enable additional, novel applications based on the general knockoff strategy described here.

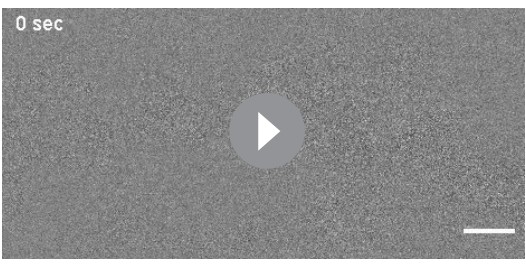

**Video 3.** Acute knockoff of SYT1 unclamps spontaneous release a, Larger area from the same image sequence in *Video 2*. Numerous spontaneous release events are seen throughout the FOV. Scale bar, 5 μm.

https://elifesciences.org/articles/56469#video3

We applied knockoff to SYT1, the Ca$^{2+}$ sensor for synchronous neurotransmitter release, for two specific reasons. First, as discussed above, the function of SYT1 in triggering and synchronizing neurotransmitter release is well documented, and thus provides a robust read-out to monitor the efficacy of knockoff. Second, the hypothetical clamping function of SYT1 remains the subject of debate, and a consensus regarding this important question has not been reached. Using S1-SELF, we were able to monitor not only the loss of synchronous neurotransmitter release, we were also able to measure the rate of miniature (mGT) release as SYT1 was acutely degraded. This work revealed a strong correlation between loss of synchronous release and an increase in rate of spontaneous release

(*Figure 6h*). This correlation suggests that a single molecule, SYT1, directly controls both processes (*Bai et al., 2016*). Importantly, these changes occur on timescales that aremuch faster than homeostatic plasticity mechanisms (hours versus days). In the course of these experiments, we also demonstrate the utility of the fluorescent glutamate sensor iGluSnFR for monitoring different modes of glutamate release in dissociated hippocampal cultures. Here, using peak $\Delta F/F_0$ iGluSnFR fluorescence, we were able to measure synchronous (peaks $\leq$ 10 ms post stimulus), asynchronous (peaks > 10 ms post stimulus), and spontaneous glutamate release in dissociated WT and S1KO hippocampal neurons. Thus, the combination of iGluSnFR and knockoff provide a powerful new means to interrogate the function of presynaptic proteins involved in release.

In summary, this study demonstrates the utility of a new, rapid, efficient, protein disruption method. We used this technique to address the unresolved question of whether SYT1 has a second function as a fusion clamp, an issue that has been debated for more than two decades (*Chapman, 2008*). Our results provide a strong argument for the dual function of SYT1, as a $Ca^{2+}$ sensor that triggers rapid evoked synchronous release, and as a clamp that inhibits spontaneous release under resting conditions (*Bai et al., 2016*). Knockoff was critical in answering this question as neurons express numerous long-lived proteins, and synaptic transmission is notoriously adaptive (homeostatic plasticity). Importantly, knockoff is also reversible; simply re-adding the inhibitor to culture media will protect newly translated copies of the targeted protein. Hence, proteins can be disrupted during different phases of neuronal development. Additionally, knockoff can be used to disrupt any terminal domain of a protein, there is no a priori need to disrupt the entire protein. However, a caveat of knockoff is that the N-terminal fragment is not rapidly degraded. Such fragments may persist for some time, causing off target effects, including dominant-negative activity, so this should be taken into consideration. We also note that knockoff cleavage sites could even be inserted into intramembrane loops and used to separate domains of polytopic membrane proteins. Given the short length of the cleavage site (10 residues), the relatively small size of the NS3/4 protease, and robust protease activity, knockoff can potentially be applied to myriad proteins. An exciting future direction will focus on testing knockoff in vivo. Application of knockoff in vivo will primarily rely on the pharmacokinetics of paritaprevir. In early clinical testing, high doses of PRV were well tolerated in mice and, in humans, PRV reaches micromolar concentrations in serum (*Menon et al., 2017*).

## Materials and methods

### Ethics statement

All animal care and experiment protocols in this study were conducted under the guidelines set by the NIH Guide for the Care and Use of Laboratory Animals handbook. The protocols were reviewed and approved by the Animal Care and Use Committee (ACUC) at the University of Wisconsin, Madison (Laboratory Animal Welfare Public Health Service Assurance Number: A3368-01).

### Cell culture

Sprague Dawley rat hippocampal and cortical neurons were isolated at E18 (Envigo). Mouse hippocampal neurons from the *Syt1* floxed mouse strain (*Quadros et al., 2017*) were isolated at P0. Hippocampal and cortical tissue was dissected and maintained in chilled hibernate A media (BrainBits; HA). Neuronal tissue was then incubated with 0.25% trypsin (Corning; 25–053 CI) for 25 min at 37°C, washed with DMEM (Thermofisher; 11965–118) supplemented with 10% fetal bovine serum (FBS, Atlanta Biological) plus Penicillin-Streptomycin (pen/strep) (Thermofisher; MT-30–001 CI), and triturated. Rat neurons were plated on 18 mm glass coverslips (Warner instruments; 64–0734 [CS-18R17]) coated with poly-D-lysine (Thermofisher; ICN10269491). Mouse neurons were plated similarly but with the addition of EHS laminin (Thermofisher; 23017015). Neurons were initially plated in DMEM with 10% FBS and pen/strep for 1 hr. Then, the media was replaced with Neurobasal-A (Thermofisher; 10888–022) medium supplemented with B-27 (2% Thermofisher; 17504001), Glutamax (2 mM Gibco; 35050061), and pen/strep. Neurons were used between 14–21 DIV. HEK293T cells (ATCC) were cultured in DMEM supplemented with 10% FBS and pen/strep, passaged upon 80% confluency. HEK293T cells were tested for mycoplasma contamination using the Universal Mycoplasma Detection Kit (ATCC; 30–1012K), and validated as HEK293T cells using Short Tandem Repeat profiling by ATCC (ATCC; 135-XV).

## Lentivirus production and use

For lentiviral use, relevant constructs were subcloned into a FUGW transfer plasmid (FUGW was a gift from David Baltimore [Addgene plasmid # 14883; http://n2t.net/addgene:14883; RRID:Addgene_14883]) (*Lois et al., 2002*). This plasmid was modified to replace the ubiquitin promoter with the human synapsin I promoter (*Kügler et al., 2003*). Lentiviral particles were generated by co-transfecting pF(UG) transfer plasmid with the packaging plasmids, pCD/NL-BH*DDD and pLTR-G into HEK293T cells (pCD/NL-BH*DDD was a gift from Jakob Reiser [Addgene plasmid # 17531; http://n2t.net/addgene:17531; RRID:Addgene_17531]), (pLTR-G was a gift from Jakob Reiser [Addgene plasmid # 17532; http://n2t.net/addgene:17532; RRID:Addgene_17532]) (*Zhang et al., 2004*). HEK293T cells were grown in DMEM supplemented with 10% FBS and pen/strep. Following transfection, the supernatant was collected after 48–72 hr of expression, passed through a 0.45 mm PVDF filter, and concentrated by ultra-centrifugation at 110,000 x g for two hours. Viral particles were re-suspended in $Ca^{2+}$/$Mg^{2+}$-free phosphate buffered saline (PBS) and kept at −80°C until use (*Kutner et al., 2009*). iGluSnFR lentivirus was added to neurons just after plating (0 DIV). SELF-mito lentivirus was added at 2–6 DIV and S1-SELF lentivirus at 1–2 DIV. S1-SELF lentivirus was titrated to express at the same levels as the WT protein in rat cortical neurons via immunoblot analysis for each new production batch. All other lentiviruses were titrated based on fluorescence.

## Live-cell, non iGluSnFR, imaging

HEK293T cells transfected with the knockoff model and primary neuronal cultures transduced with the model were imaged using the Zeiss LSM880 with Airyscan confocal microscope. Mitochondria were counterstained with MitoTracker Red CMXRos (ThermoFisher) and MitoTracker Green FM (ThermoFisher). Coverslips containing HEK293T were placed in standard imaging media (ECF (extracellular fluid/ECF) consisted of 140 mM NaCl, 5 mM KCl, 2 mM $CaCl_2$, 2 mM $MgCl_2$, 5.5 mM glucose, 20 mM HEPES (pH 7.3), B27 (Gibco), glutamax (Gibco), and positioned on the microscope. For long term neuron knockoff imaging, conditioned neurobasal growth media was used during imaging and $CO_2$ controlled via Tokai incubation chamber. For both types of cells, images were acquired at 34°C, temperature and humidity were controlled using a tokai incubation chamber. Cells were imaged in Fast Airyscan mode and processed with automatic Airyscan deconvolution settings after image acquisition.

## Knockoff protocol

The general knockoff protocol is outlined in *Figure 3A*. Specifically, after introduction of the NS3/4A containing vectors, protease inhibitor was added to the media at the desired concentration. All inhibitors tested appeared stable for the duration of the experiments. Fresh media, supplements, and inhibitor (NS3/4A protease inhibitor), were added to neuronal cultures every 3–4 days. Separately, additional neuronal cultures were grown and used as a source of conditioned media. For inhibitor washout, neurons were washed twice with HEPES buffered ECF, then allowed to incubate in fresh ECF for 5 min. After 5 min, media was aspirated and conditioned media was applied to the neuronal culture for the duration of the experiment (i.e. 2, 4, 6, or 8 hr).

## Immunoblot analysis

Immunoblots were performed using standard procedures and PVDF membranes (Immobilon-FL; EMD Millipore). Briefly, total protein was collected from dissociated neuronal cultures using 150 μl lysis buffer containing 1x PBS, 2% SDS, 1% Triton x-100, and 10 mM EDTA, plus the following protease inhibitors: 2 mM PMSF, aprotinin, leupeptin, and pepstatin A. Samples were then incubated at 100°C for 5 min after addition of 50 μl sample buffer (SB) (DTT, glycerol, and bromophenol blue). For protein detection, 10–20 μl of protein lysate was subjected to SDS-PAGE, using 10% or 13% gels, and transferred to a PVDF membrane. After the transfer of proteins, the gel was incubated with (4.5:4.5:1) mix of water:methanol:trichloroethanol) (TCE) for 5 min. The TCE was activated in the gel by exposure to UV light (300 nm) for 5 min. Activated TCE (cross linked proteins) were detected by 2.0 s exposure to 300 nm illumination and used as a protein load control (*Ladner et al., 2004*).

After transfer, the PVDF membrane was blocked with 5% nonfat milk protein in Tris-buffered saline plus 1% Tween20 (TBST) for 30 min and incubated with primary antibody overnight. The

membrane was then washed three times and incubated with a secondary antibody. Primary antibodies were: anti-GFP (1:1000, 7.1 and 13.1) (Roche; #11 814 460 001; RRID:AB_390913), anti-SYT1 (1:500, 48) (DSHB; #mAB 48; RRID:AB_2199314), anti-NS3 (1:500, 1B6) (VWR; 75792–858; RRID:AB_732840), anti-CDK5 (1:500, DC34) (Thermofisher; AHZ0492; RRID:AB_2536380), anti-CRE (1:500, 7.23) (Abcam; ab24607; RRID:AB_448179), anti-Syp (1:1000) (SySy; 101 004; RRID:AB_1210382), anti-Syb2 (1:1000) (SySy; 104 211C5, RRID:AB_2619757), anti-Gyr1 (1:1000) (SySy; 103 002, RRID:AB_887818), anti-FLAG (1:1000) (Sigma; F3165, RRID:AB_259529). Secondary antibodies were: goat anti-mouse IgG-HRP (Biorad, 1706516; RRID:AB_11125547), goat anti-mouse IgG2b-HRP (Biorad, M32407; RRID:AB_2536647), and goat anti-rabbit IgG-HRP (Biorad, 1706515, RRID:AB_11125142). Immunoblots were imaged using Luminata Forte Western HRP substrate (EMD Millipore) and the Amersham Imager 680 imaging system (GE Healthcare). Immunoreactive bands were analyzed by densitometry and contrast was linearly adjusted for publication using Fiji (*Schindelin et al., 2012*). SELF-mito immunoblots were analyzed after background subtraction using a rolling ball radius of 20 pixels.

## Immunocytochemistry (ICC)

Dissociated mouse hippocampal cultures were fixed with 4% paraformaldehyde in PBS, at 37°C, for 15 min. Fixed neurons were washed twice with PBS and then permeabilized and quenched, at RT, for 10 min in PBS with 0.2% saponin and 50 mM ammonium chloride. Samples were then incubated for one hour in blocking buffer 1 (PBS with 10% goat serum, 10% bovine serum albumin (BSA), 0.02% sodium azide ($NaN_3$), and 0.02% saponin), followed by a 1 hr incubation with primary antibodies diluted in blocking buffer 2 (PBS with 1% BSA, 0.02% $NaN_3$, and 0.02% saponin). Primary antibodies were: anti-CRE (1:100, 2D8) (Millipore; MAB3120, RRID:AB_2085748), anti-MAP2 (1:100) (Millipore; AB15452, RRID:AB_805385), anti-MAP2 (1:500) (Millipore; AB5543, RRID:AB_571049), anti-PSD95 (1:250) (ThermoFisher; MA1-046, RRID:AB_2092361), anti-vGlut1 (1:500) (Millipore; AB5905, RRID:AB_2301751), anti-SYT1 C2A (1:50, 48) (DSHB; mAB 48; RRID:AB_2199314), anti-SYT1 luminal domain (LD) (1:200) (SySy; 105 103; RRID:AB_11042457), and anti-synaptophysin (SYP) (1:500) (SySy; 101 004; RRID:AB_1210382). Samples were then washed four times in wash buffer (PBS with 0.02% $NaN_3$, and 0.02% saponin) and incubated for one hour in secondary antibody diluted in blocking buffer 2. Secondary antibodies used include, goat anti-chicken IgY-Alexa Fluor 405 (1:500) (abcam; ab175675, RRID:AB_2810980), goat anti-mouse IgG1 IgG-Alexa Fluor 488 (1:500) (Thermofisher; A-21121, RRID:AB_2535764), goat anti-guinea pig IgG-Alexa Fluor 488 (1:500) (Thermofisher; A-11073; RRID:AB_2534117), goat anti-rabbit IgG-Alexa Fluor 546 (1:500) (Thermofisher; A-11010, RRID:AB_2534077), goat anti-chicken IgG-Alexa Fluor 546 (1:500) (Thermofisher; A-11040, RRID:AB_2534097), goat anti-guinea pig IgG-Alexa Fluor 546 (1:500) (Thermofisher; A-11074, RRID:AB_2534118), goat anti-mouse IgG1 IgG-Alexa Fluor 647 (1:500) (Thermofisher; A-21240, RRID:AB_2535809), and goat anti-mouse IgG2b-Alexa Fluor 647 (1:500) (Thermofisher; A-21242; RRID:AB_2535811). After secondary antibody incubation, neurons were washed twice in wash buffer and three times in PBS before being mounted on glass slides with ProLong Diamond Antifade Mountant (Thermfisher; P36965). Images for *Figure 1* were acquired on a Zeiss LSM 880 with a 63 × 1.4 NA oil immersion objective in confocal mode. Images for *Figure 4* were acquired on the LSM 880 but using the Airyscan super-resolution detector; identical laser and gain settings were used for all samples. Images were deconvolved using automatic Airyscan settings. The same linear brightness and contrast adjustments were applied to all ICC images for publication.

## iGluSnFR imaging

Live cell fluorescence images were acquired on an Olympus IX83 inverted microscope equipped with a cellTIRF 4Line excitation system using an Olympus 60x/1.49 Apo N objective and an Orca Flash4.0 CMOS camera (Hamamatsu Photonics) running Metamorph software that was modified to run concurrently with Olympus 7.8.6.0 acquisition software from Molecular devices. Standard imaging media, extracellular fluid (ECF) was used for iGluSnFR experiments. For stimulated release experiments, unless otherwise noted, $CaCl_2$ in ECF, was lowered to 1 mM, and D-AP5 (50 μM) (Abcam; ab120003), CNQX (20 μM) (Abcam; ab120044), Picrotoxin (100 μM) (Tocris; 1128) were added to the imaging media. For spontaneous release experiments, 2 mM $CaCl_2$ was used; and in place of D-AP5, CNQX, and Picrotoxin; TTX (1 μM) (Abcam; ab120055) was added to the imaging

media. Single image planes were acquired with 10 ms exposure (100 Hz) using 488 nm excitation and 520 emission. Single planes were selected so that the highest amount of iGluSnFR expressing dendrite arbor was in focus. Laser intensity was set so that the amplitude of iGluSnFR peaks ($\Delta F/F_0$) from a 0.5 Hz field stimulus did not change over the course of at least 20 s (photobleaching error). For single stimuli imaging, 150 frames (10 ms exposure/1.5 s total) were collected and a single field stimulus was triggered at 500 ms after the initial frame. For train stimuli imaging, 200 frames (10 ms exposure/2 s total) were collected and a train of 5 stimuli were triggered at 10 Hz, starting at 500 ms after the initial frame. For spontaneous imaging, 300 frames (200 ms exposure/60 s total) were collected. Single and train (10 Hz) stimuli were triggered by a Grass SD9 stimulator through platinum parallel wires attached to a field stimulation chamber (Warner Instruments; RC-49MFSH). Voltage for field stimulus was set to the lowest voltage that reliably produced presynaptic $Ca^{2+}$ transients in all (>95%) presynaptic boutons using synaptophysin-GCaMP6f as a presynaptic $Ca^{2+}$ reporter. All iGluSnFR imaging experiments were performed at 33–34℃. Temperature and humidity were controlled during experiments by a Tokai incubation controller and chamber.

## iGluSnFR quantification

Image series were acquired as described under the iGluSnFR imaging section. Image sets were opened, ROIs created, and fluorescence intensity measured using ImageJ (Fiji) (*Schindelin et al., 2012*). ROIs were created based on a custom workflow to identify changes in fluorescence of the selected image series. For stimulated release imaging, the average projection (created using the pre-stimulus fluorescence baseline) was subtracted from the maximum projection (created using the entire image series). This result was duplicated (Duplicate result) and mean filtered (Duplicate filtered) using a rolling ball radius of 10 pixels. These images were subtracted and the result was used to threshold changes in fluorescence. After thresholding, images were made binary and a watershed function was run to separate objects that touched each other. Using this procedure, objects (>10 pixels) were created and ROIs were defined. These ROIs were used to measure fluorescence changes over time from the original image series. The results were copied and imported into Axograph X 1.7.2 (Axograph Scientific) where traces were normalized, and background subtracted using pre-stimulus data. For miniature neurotransmitter release ROI selection, a walking average image series was created using the original image series as a template, then the walking average image series was subtracted from the original image series. From here, spontaneous events were manually identified and counted. For ROI analysis, ROIs were manually drawn around spontaneous events and for background ROIs, these same ROIs were moved off center to an area that was not counted as an event.

## Stimulated release macro

```
run("Duplicate...", "title=OG duplicate");
run("Z Project...", "stop=45 projection=[Average Intensity]");
selectWindow("OG");
run("Z Project...", "projection=[Max Intensity]");
imageCalculator("Subtract create 32-bit", "MAX_OG","AVG_OG");
selectWindow("Result of MAX_OG");
run("Duplicate...", "title=Duplicate result");
run("32-bit");
run("Duplicate...", "title=Duplicate filtered");
selectWindow("Filtered");
run("Mean...", "radius=" +10);
run("Image Calculator...", "image1=Duplicate result operation=Subtract image2=-
Duplicate filtered create");
run("Rename...", "title=Result");
run("Threshold...");
setAutoThreshold("Default dark");
waitForUser("Adjust threshold – Don't mess up ", "Hit OK after setting threshold")
```

```
run("NaN Background");
setOption("BlackBackground", false);
run("Make Binary");
run("Watershed");
run("Analyze Particles...", "size=10 Infinity display exclude clear include sum-
marize add");
selectWindow("Duplicate filtered");
close();
selectWindow("Duplicate result");
close();
selectWindow("Result of MAX_OG");
close();
selectWindow("MAX_OG");
close();
selectWindow("AVG_OG");
close();
 if (isOpen("Intensity")) {
   selectWindow("Intensity");
   run("Close"); };
 if (isOpen("Area")) {
   selectWindow("Area");
   run("Close"); };
run("Set Measurements...", "mean redirect=None decimal=1");
selectWindow("OG");
roiManager("Show None");
roiManager("SelectAll");
roiManager("Multi Measure");
String.copyResults();
IJ.renameResults("Intensity");
run("Set Measurements...", "area mean min median skewness redirect=None deci-
mal=1");
roiManager("SelectAll");
roiManager("Measure");
IJ.renameResults("Area");
selectWindow("OG"); close();
```

## Spontaneous release macro

```
run("Duplicate...", "title=OG duplicate");
run("WalkingAverage ");
selectWindow("OG");
run("Delete Slice");
run("Delete Slice");
run("Delete Slice");
imageCalculator("Subtract create 32-bit stack", "OG","walkAv");
selectWindow("OG");
close();
selectWindow("walkAv");
close();
```

## Electrophysiology

Soma voltage-clamp recordings were carried out using a Multiclamp 700B amplifier (Molecular Devices) acquired using a Digidata 1440A (Molecular Devices) and Clampex 10 software (Molecular Devices) recorded at 10 kHz. Patch pipettes (3–5 MΩ) were pulled from borosilicate glass (Sutter Instruments). Neurons were held at −70 mV while series resistance was compensated, and recordings were discarded if the access resistance rose above 15 MΩ by the end of the recording. Tetrodotoxin (TTX, 1 mM, Abcam), D-AP5 (50 mM, Abcam), and picrotoxin (100 mM, Abcam) were included in the bath solution to isolate miniature excitatory events. Recordings of mouse hippocampal neurons at 14–16 DIV were carried out at 34°C in a bath solution containing: 128 mM NaCl, 5 mM KCl, 2 mM CaCl$_2$, 1 mM MgCl$_2$, 30 mM glucose, and 25 mM HEPES, pH 7.3 at 305 mOsm. Internal solution contained: 130 mM K-Gluconate, 1 mM EGTA, 10 mM HEPES, 2 mM ATP, 0.3 mM GTP, and 5 mM NA$_2$phosphocreatine, pH 7.35 at 275 mOsm. Recorded traces were analyzed using Clampfit 10 (Molecular Devices) and down sampled for figure presentation to 2 kHz using Axograph X 1.7.2 (Axograph Scientific).

## Colocalization quantification

Pearson's correlation coefficient (PCC) was measured using Just Another Colocalization Plugin (JACoP) (*Bolte and Cordelières, 2006*) for Fiji. Briefly, neurons were grown, stained, and imaged using the procedures that are detailed above. Five to six rectangular regions of interest were randomly selected, and PCC analyzed using costes automatic threshold in JACoP.

## Live dead assay

The Dead Cell Apoptosis Kit with Annexin V FITC and PI (Thermofisher; V13242) kit was used to determine toxicity of various HCV protease inhibitors (Ciluprevir/CLV, Asunaprevir/ASV, Simeprevir/ SMV, Danoprevir/DNV, Paritaprevir/PRV, Grazoprevir/GZV, Glecaprevir/GCV) and performed according to the manufacturer protocol. Briefly, HEK293T cells were grown in the presence of various HCV protease inhibitors for one day. Cultures were then incubated with a mix of annexin V-FITC, propidium iodide (PI), and Hoechst 33342 (Thermofisher; H3570) for 30 min. Samples were then immediately imaged using widefield microscopy, using standard blue (Hoechst), green (annexin V), and red (PI) filter sets. Using this protocol, all cells stain blue with Hoechst; necrotic cells will stain blue and red (PI), apoptotic cells will stain blue and green (annexin V), and dead cells will stain blue, red, and green (PI and annexin V). The total number of cells was determined using Hoechst positive puncta. Absolute cell numbers in each trial were normalized to control. Inhibitor toxicity was also assayed using primary rat cortical cultures. For neurons, the ratio of PI+ puncta (Dead) to Hoechst+ puncta (Total) was calculated.

## Plasmid construction

The lentivirus used for CRE-mediated excision of *Syt1* exon five from floxed mice (*Quadros et al., 2017*), was made using the above lentivirus method and using the transfer plasmid pLenti-hSynapsin-CRE-WPRE (pLenti-hSynapsin-CRE-WPRE was a gift from Fan Wang [Addgene plasmid # 86641; http://n2t.net/addgene:86641; RRID:Addgene_86641]) (*Sakurai et al., 2016*). The glutamate sensor, iGluSnFR, was PCR amplified using the iGluSnFR template (with super-folding mutations) (pAAV.hSynapsin.SF-iGluSnFR.A184V was a gift from Loren Looger [Addgene plasmid # 106175; http://n2t.net/addgene:106175; RRID:Addgene_106175]) (*Marvin et al., 2018*), and subcloned into our lentivirus transfer plasmid (CamKII promoter) after the addition of a Golgi export sequence (*Parmar et al., 2014*), and an ER exit motif (*Stockklausner et al., 2001*), to the carboxy terminus. These export motifs improved trafficking of the sensor to the plasma membrane. The knockoff model (SELF-mito) was created by PCR Splicing by Overlap Extension (SOE) the NS3/4A protease (*Butko et al., 2012*), monomeric-superfolder green fluorescent protein (msGFP - from lab stock), the NS3/4A substrate site (EDVVCC/SMSY), and the mitochondrial targeting signal from OMP25 (*Nemoto, 1999*). Flexible GS(GSS)$_4$ linkers were added in between the protease and msGFP, and msGFP and the cleavage site. Knockoff models with alternate cleavage sites were created using a long, 3', oligonucleotide primer encoding the alternate cleavage site. The knockoff model (SELF-mito) was subcloned into pEF-GFP (for HEK293T expression) and FUGW (for neuronal expression) (pEF-GFP was a gift from Connie Cepko [Addgene plasmid # 11154; http://n2t.net/addgene:11154; RRID:Addgene_11154])

(*Matsuda and Cepko, 2004*). The cleavable SYT1 (S1-SELF) was created using the rat cDNA from previous lab constructs. The substrate cleavage site (ADVVCC/QMSY) was inserted after amino acid 100 using PCR SOE techniques. Furthermore, the codon optimized NS3/4A (custom geneblock from IDT), was also added to this same construct using PCR SOE, along with a long (67 amino acid) flexible linker composed of primarily of glycine, serine, and alanine at the 5' end of the protease (3' of *Syt1*), and a short FLAG tag at the 3' end of the protease. The cleavable Synaptophysin (SYP1-SELF) was created by inserting the cleavage site into the carboxy terminus at amino acid number 234, just prior to the YG(P/Q) repeats (*Südhof et al., 1987*) that have been shown to control rates of endocytosis (*Kwon and Chapman, 2011*). This was then amended with the same codon optimized protease tag that was used for S1-SELF. These constructs were all subcloned into our modified FUGW vector described under lentivirus production and use.

## Compounds and chemicals

HCV NS3 protease inhibitors were obtained from the following manufacturers: Ciluprevir/CLV (AffixScientific; 300832-84-2), Simeprevir/SMV (Apexbio; A3820), Asunaprevir/ASV (Apexbio; A3195), Danoprevir/DNV (Apexbio; A4024), Paritaprevir/PRV (MedChemExpress/MCE; HY-12594), Grazoprevir/GZV (MedChemExpress/MCE; HY-152980), Glecaprevir/GCV (MedChemExpress/MCE; HY-17634). All other materials, if not stated, were obtained from Sigma-Aldrich.

## Statistics

All values from quantification and the number of trials (n) in each experiment are listed in the Figure Legends. Summary statistics for *Figure 5b* (*Figure 5—source data 1*), *Figure 6a* (*Figure 6—source data 1*), *Figure 6f* (*Figure 6—source data 2*) *Figure 6g* (*Figure 6—source data 1*) were obtained by performing multiple comparisons using the non-parametric Kruskal-Wallis test with Dunn's multiple comparison correction. Single exponential functions were fitted to the data in *Figure 2f* and *Figure 3c*. Linear regression was performed on the data from *Figure 6h*. Asterisks correspond to p-values as follows: *; $p \leq 0.05$, **; $p \leq 0.005$, ***; $p \leq 0.001$. Expected sample sizes were not estimated or predetermined. All statistical analyses were conducted using GraphPad Prism 7.04 (GraphPad Software Inc).

## Acknowledgements

This study was supported by grants from the NIH (MH061876 and NS097362 to ERC). JDV was supported by a postdoctoral fellowship from the National Institutes of Health F32 NS098604. ERC is an Investigator of the Howard Hughes Medical Institute.

## Additional information

### Funding

| Funder | Grant reference number | Author |
| --- | --- | --- |
| National Institutes of Health | MH061876 | Edwin R Chapman |
| National Institutes of Health | NS097362 | Edwin R Chapman |
| Howard Hughes Medical Institute | | Edwin R Chapman |
| National Institutes of Health | NS098604 | Jason D Vevea |

The funders had no role in study design, data collection and interpretation, or the decision to submit the work for publication.

### Author contributions

Jason D Vevea, Conceptualization, Data curation, Formal analysis, Funding acquisition, Investigation, Visualization, Methodology, Writing - original draft, Writing - review and editing; Edwin R Chapman, Resources, Supervision, Project administration, Writing - review and editing

## Author ORCIDs

Jason D Vevea (iD) https://orcid.org/0000-0002-3068-973X
Edwin R Chapman (iD) https://orcid.org/0000-0001-9787-8140

## Ethics

Animal experimentation: All animal care and experiment protocols in this study were conducted under the guidelines set by the NIH Guide for the Care and Use of Laboratory Animals handbook. The protocols were reviewed and approved by the Animal Care and Use Committee (ACUC) at the University of Wisconsin, Madison (Laboratory Animal Welfare Public Health Service Assurance Number: A3368-01).

## Decision letter and Author response

Decision letter https://doi.org/10.7554/eLife.56469.sa1
Author response https://doi.org/10.7554/eLife.56469.sa2

# Additional files

## Supplementary files

• Supplementary file 1. Knockoff development using HEK293T cells and model substrates. Table summarizing commercially available HCV inhibitors and their properties.

• Transparent reporting form

## Data availability

All data generated or analysed during this study are included in the manuscript and supporting files.

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
