## [Decision Letter]

**Acceptance summary:**

The technology that you describe here could be very useful in numerous contexts for many biologists with many different interests and we look forward to its application in different species as well.

**Decision letter after peer review:**

Thank you for submitting your article "Knockoff: Acute disruption of the synaptic vesicle membrane protein synaptotagmin 1" for consideration by *eLife*. Your article has been reviewed by Kenton Swartz as the Senior Editor, a Reviewing Editor, and two reviewers. The following individuals involved in review of your submission have agreed to reveal their identity: J Troy Littleton (Reviewer #1).

The reviewers have discussed the reviews with one another and the Reviewing Editor has drafted this decision to help you prepare a revised submission.

We would like to draw your attention to changes in our revision policy that we have made in response to COVID-19 (https://elifesciences.org/articles/57162). Specifically, when editors judge that a submitted work as a whole belongs in *eLife* but that some conclusions require a modest amount of additional new data, as they do with your paper, we are asking that the manuscript be revised to either limit claims to those supported by data in hand, or to explicitly state that the relevant conclusions require additional supporting data. We did not edit or summarize the reviews and you will notice that there is a major concern related to the novelty of the strategy and that concern should be fully addressed and the references that reviewer 2 lists should be mentioned and discussed.

Reviewer #1:

The manuscript by Vevea and Chapman describes a new method (termed knockoff) to acutely reduce protein function of transmembrane proteins by engineering a construct encoding the druggable HCV NS3/4a protease and a NS3/4 cleavage site into the cytoplasmic portion of the protein of interest. Following removal of the protease inhibitor, the target protein is cleaved and the cytosolic fragment undergoes N-end degradation due to an exposed Q amino acid on the new N-terminus. The authors demonstrate the approach works in hippocampal cultures using the synaptic vesicle proteins Synaptotagmin 1 (Syt1) and Synaptophysin 1. The authors also compare it to several other published approaches using degradation as a tool and show it performs superiorly. As such, the method represents an exciting addition for those interested in acutely (2-8 hours) disrupting a transmembrane protein.

The authors follow-up on their method development to characterize acute disruptions of Syt1 using the tool. They provide some new iGluSnFR improvements for imaging synaptic transmission in mammalian neurons that will be welcomed in the field. They show convincing data that with both physiology and imaging that similar to CRE-dependent removal of Syt1, degradation of Syt1 in hippocampal cultures reduce synchronous synaptic vesicle fusion and increase spontaneous release. They emphasize their acute approach allows them to exclude long-term compensation as a mechanism for the increased mini frequency (clamp model), arguing for the likelihood that Syt1 indeed functions as both an activator of synchronous fusion and an inhibitor of spontaneous release, as proposed in other studies.

Overall, it's a nice set of experiments with some valuable toolkits for the neuroscience field, and potentially more broadly for the cell biology community in general.

A few comments:

1) Have the authors tried the approach in vivo? Can the drug be administered to mice to block the protease, and then removed to have the protein cleaved in vivo? This would be a very powerful addition to the toolkit if the method could be used outside of cell culture. It's not an essential experiment, but would be nice to know if the authors tried it. Given synaptophysin is not an essential gene, that seems like an ideal system to test it. Alternatively, it could be done with the Syt1 line to see if the mice develop neurological symptoms following removal of the drug. A western analysis could show if Syt1 were cleaved in vivo. It would be a pity if the approach can only be used in cell culture.

2) It is worth having a paragraph on the caveats of the method. Although I really like the clever toolkit, one could imagine a few issues depending upon the protein being characterized:

A) The authors show the residual transmembrane fragment is long-lived. That might cause some dominant-negative effects compared to a null mutant depending on the protein.

B) The Syt1 degradation looks like it could be non-complete. Figure 4C shows the protein levels don't drop to the same as the CRE null situation. Supplementary file 5 indicates evoked release is not reduced as strongly as the null (Control: ~68, Syt1 KO: ~6; Syt1-SELF 8h ~13). Not a big deal, but there is likely to be a small fraction that escapes cleavage. Again, depending on the protein, 5-10% might have significant function for its biology.

C) What would happen over longer-term expression without the inhibitor. Does cleavage start to happen very early before the protein is actually even on synaptic vesicles, for example, following entry into the ER or on vesicles in transit. Did the authors see any hints of what is happening to Syt1 in the cell body of their long-term 8 hour experiments? Does is still co-localize with other SV proteins during axonal transport in the 8 hour experiments?

3) I would recommend changing the wording of the + PRV inhibitor experiments. The authors refer to this condition as "rescue", when it should be called "blocked" or another similar term. Rescue has a specific genetic connotation where a mutant phenotype is reverted by putting back the wildtype gene. Better to avoid that confusion by using "blocked" as the term with inhibitor.

Reviewer #2:

In this manuscript Vevea and Chapman developed a knock-out method that depends on depends on NS3/4A cis cleavage of targeted protein that leads to N-end rule mediated degradation of targeted protein and provide proof of principle experiments on integral membrane protein Synaptotagmin 1 (Syt1) and Synaptophysin 1. The authors start by showing that Syt1 has a highly perduring pool that limits the success of genetic knock-out approaches in mature mouse neurons. Moreover, previously established AID mediated knock-down of Syt1 is leaky, limiting temporal control of knock-down. The authors use NS3/4 mediated cleavage of Syt1 as a means to rapidly knock-out syt1. They design and optimize a cis cleavage site and tag syt1 with the cleavage site of NS3/4 and the protease that targets it in the absence of NS3/4 inhibitors. They test and optimize the inhibitor levels and show that the inhibitor PRV is not toxic at varying levels and at low levels it inhibits the cleavage of an optimized (for drug mediated inhibition) target site of NS3/4A. Wash out of the inhibitor leads to cleavage of the cis target site, and N-end rule mediated degradation of the target gene near completion at 166 minutes after inhibitor wash off at mouse hippocampus neuronal cultures. Finally, they show that acute knock-out of syt1 increases spontaneous mini frequency, strengthening the hypothesis that syt1 functions as a clamp against spontaneous vesicle release.

The technique works and the results are compelling, but the authors fail to refer to and discuss key publications that severely limit the novelty of their paper. Indeed, the NS3/4 mediated cis cleavage is used in SMASh tag (Chung et al., 2015) where the targeted protein would be tagged with NS3 protease and a degron that can be cleaved off by NS3 protease. When an inhibitor is added this cleavage is interrupted, which results in rapid degradation of the target. Moreover, SMASh tag was shown to work with transmembrane protein GluRIIA in that publication. Additionally, TEV mediated target cleavage that results in target degradation through N-end rule was published in Taxis et al., 2009. Due to these references, I believe the novelty of the method is very limited to warrant publication in *eLife*.

---

## [Author Response]

Reviewer #1:The manuscript by Vevea and Chapman describes a new method (termed knockoff) to acutely reduce protein function of transmembrane proteins by engineering a construct encoding the druggable HCV NS3/4a protease and a NS3/4 cleavage site into the cytoplasmic portion of the protein of interest. Following removal of the protease inhibitor, the target protein is cleaved and the cytosolic fragment undergoes N-end degradation due to an exposed Q amino acid on the new N-terminus. The authors demonstrate the approach works in hippocampal cultures using the synaptic vesicle proteins Synaptotagmin 1 (Syt1) and Synaptophysin 1. The authors also compare it to several other published approaches using degradation as a tool and show it performs superiorly. As such, the method represents an exciting addition for those interested in acutely (2-8 hours) disrupting a transmembrane protein.The authors follow-up on their method development to characterize acute disruptions of Syt1 using the tool. They provide some new iGluSnFR improvements for imaging synaptic transmission in mammalian neurons that will be welcomed in the field. They show convincing data that with both physiology and imaging that similar to CRE-dependent removal of Syt1, degradation of Syt1 in hippocampal cultures reduce synchronous synaptic vesicle fusion and increase spontaneous release. They emphasize their acute approach allows them to exclude long-term compensation as a mechanism for the increased mini frequency (clamp model), arguing for the likelihood that Syt1 indeed functions as both an activator of synchronous fusion and an inhibitor of spontaneous release, as proposed in other studies.Overall, it's a nice set of experiments with some valuable toolkits for the neuroscience field, and potentially more broadly for the cell biology community in general.A few comments:1) Have the authors tried the approach in vivo? Can the drug be administered to mice to block the protease, and then removed to have the protein cleaved in vivo? This would be a very powerful addition to the toolkit if the method could be used outside of cell culture. It's not an essential experiment, but would be nice to know if the authors tried it. Given synaptophysin is not an essential gene, that seems like an ideal system to test it. Alternatively, it could be done with the Syt1 line to see if the mice develop neurological symptoms following removal of the drug. A western analysis could show if Syt1 were cleaved in vivo. It would be a pity if the approach can only be used in cell culture.

This is an excellent suggestion. We have not yet tried knockoff in vivo, but we are planning these experiments. We are confident that they will work because paritaprevir (PRV) has been reported to reach μM concentrations in human serum, and has been shown to be well-tolerated in rodent toxicology studies, using does that are 100-fold greater than the doses used in humans (unlike many other NS3/4 protease inhibitors). We will need to do a full dosing curve with and without ritonavir to assess cost and ultimate feasibility, but we are optimistic about these experiments.

2) It is worth having a paragraph on the caveats of the method. Although I really like the clever toolkit, one could imagine a few issues depending upon the protein being characterized:A) The authors show the residual transmembrane fragment is long-lived. That might cause some dominant-negative effects compared to a null mutant depending on the protein.

This is a good point, thank you. We have added this caveat/clarification to the revised manuscript, and now state: “; however, a caveat of knockoff is that the N-terminal fragment is not rapidly degraded. Such fragments may persist for some time, causing off target effects, including dominant-negative activity, so this should be taken into consideration.”.

B) The Syt1 degradation looks like it could be non-complete. Figure 4C shows the protein levels don't drop to the same as the CRE null situation. Supplementary file 5 indicates evoked release is not reduced as strongly as the null (Control: ~68, Syt1 KO: ~6; Syt1-SELF 8h ~13). Not a big deal, but there is likely to be a small fraction that escapes cleavage. Again, depending on the protein, 5-10% might have significant function for its biology.

This is another good point, and it raises an interesting facet of our findings. The immunoblot from Figure 3 shows 0.4% full length syt1 at 8 hours, however some C2A immunofluorescence can be observed from the ICC in Figure 4 (same antibody, ICC not quantified), and results in a PCC above baseline. In the ICC images, the signal detected from the C2A antibody appears more diffuse at the -4, -6, and -8-hour washout time points. We reason that the C2A domain was cleaved from synaptic vesicles but had not yet been fully degraded by the cell. It is possible that this degradation step takes hours and that is the reason for above baseline signal and residual colocalization. The reviewer also notes that the synchronous fraction in the 8-hour time point is trending higher than the complete syt1 null (although not statistically different). The reviewer asks whether this trend towards more synchronous release could be caused by residual syt1. We cannot answer this definitively because there is no statistical significance. However, it is tempting to speculate that since there is also a trend toward increased spontaneous release in the acute washout conditions, that there is not a significant amount of residual syt1 (since residual syt1 would also be expected to act as a clamp); rather, we speculate that there is some interesting difference in synaptic vesicle (SV) architecture or number when SVs are formed in the presence of syt1, but then the syt1 is acutely removed. This idea will be addressed in future studies.

C) What would happen over longer-term expression without the inhibitor. Does cleavage start to happen very early before the protein is actually even on synaptic vesicles, for example, following entry into the ER or on vesicles in transit. Did the authors see any hints of what is happening to Syt1 in the cell body of their long-term 8 hour experiments? Does is still co-localize with other SV proteins during axonal transport in the 8 hour experiments?

In Figure 3B we assayed long-term removal of inhibitor and we never detected protein. Furthermore, we performed ICC once on a long term, inhibitor removed, experiment and saw no evidence for the presence of either fragment of syt1. Based on these findings we feel confident saying that cleavage happens rapidly after protease translation. As to what happens to the N-terminal fragment, it must get somehow recycled/degraded in/near the ER because it doesn’t show up on ICC (this condition is not shown in the paper due to space and because it was only assayed once). In the acute experiments, the N-terminal fragment persists longer than the C-terminal fragment, but this fragment is lost from SVs and appears as diffuse signal throughout the axon (Figure 4).

3) I would recommend changing the wording of the + PRV inhibitor experiments. The authors refer to this condition as "rescue", when it should be called "blocked" or another similar term. Rescue has a specific genetic connotation where a mutant phenotype is reverted by putting back the wildtype gene. Better to avoid that confusion by using "blocked" as the term with inhibitor.

This is another good point, thank you. We thought about this a lot and feel that ‘blocked’ might also cause some degree of confusion, so we have opted for the term “protected”, we hope that this is clear and acceptable.

Reviewer #2:In this manuscript Vevea and Chapman developed a knock-out method that depends on depends on NS3/4A cis cleavage of targeted protein that leads to N-end rule mediated degradation of targeted protein and provide proof of principle experiments on integral membrane protein Synaptotagmin 1 (Syt1) and Synaptophysin 1. The authors start by showing that Syt1 has a highly perduring pool that limits the success of genetic knock-out approaches in mature mouse neurons. Moreover, previously established AID mediated knock-down of Syt1 is leaky, limiting temporal control of knock-down. The authors use NS3/4 mediated cleavage of Syt1 as a means to rapidly knock-out syt1. They design and optimize a cis cleavage site and tag syt1 with the cleavage site of NS3/4 and the protease that targets it in the absence of NS3/4 inhibitors. They test and optimize the inhibitor levels and show that the inhibitor PRV is not toxic at varying levels and at low levels it inhibits the cleavage of an optimized (for drug mediated inhibition) target site of NS3/4A. Wash out of the inhibitor leads to cleavage of the cis target site, and N-end rule mediated degradation of the target gene near completion at 166 minutes after inhibitor wash off at mouse hippocampus neuronal cultures. Finally, they show that acute knock-out of syt1 increases spontaneous mini frequency, strengthening the hypothesis that syt1 functions as a clamp against spontaneous vesicle release.The technique works and the results are compelling, but the authors fail to refer to and discuss key publications that severely limit the novelty of their paper. Indeed, the NS3/4 mediated cis cleavage is used in SMASh tag (Chung et al., 2015) where the targeted protein would be tagged with NS3 protease and a degron that can be cleaved off by NS3 protease. When an inhibitor is added this cleavage is interrupted, which results in rapid degradation of the target. Moreover, SMASh tag was shown to work with transmembrane protein GluRIIA in that publication. Additionally, TEV mediated target cleavage that results in target degradation through N-end rule was published in Taxis et al., 2009. Due to these references, I believe the novelty of the method is very limited to warrant publication in eLife.

This reviewer brings up an important point. We failed to make clear how knockoff is different, outperforms, and offers the ability to conduct previously impossible experiments versus these previous methods. As this reviewer cites, cell biologists have long sought after a method to acutely disrupt their protein of interest, as evidenced by these two publications. However, the utility of these previous attempts has been severely limited. We respectfully argue that, in our manuscript, we have finally worked out a degradation method that: (1) targets the protein of interest on its terminal organelle, (2) works quickly, and (3) has no discernable off target effects (no off target cleavage and no degradation of the whole organelle).

The SMASh tag is a tool for shutting off expression of a protein of interest. This method works well for turning expression off, but once the protein is made, it is no longer under the influence of the tool. This tool works by using the NS3/4 protease to control the a novel degron appended to the protein of interest. In the absence of inhibitor, the degron is cleaved from the protein of interest and the protein is produced. Addition of inhibitor prevents degron removal via protease cleavage and therefore results in degradation shortly after translation. This is important because while this method works on transmembrane proteins, it does so while the protein is presumably in the ER where it can be retrotranslocated for degradation. Furthermore, because this method works on only newly translated protein, the SMASh method will only decrease the amount of a targeted protein that either: a) has a short half-life, or b) is expressed in rapidly dividing cells like yeast or HEK cells. It will not meaningfully decrease the amount of long-lived presynaptic proteins like syt1. Conversely, Knockoff targets the mature protein on its native organelle. It is completely different and only comparable in that knockoff also uses a protease.

TEV protease has been used in cells to probe the topology of targeted proteins but we feel that it has proven to be a poor choice for studying protein disruption. To be clear, TEV sites can be engineered into proteins of interest and they can be targeted for degradation, however, TEV expression must be controlled via a controlled promoter as it does not have a small molecule inhibitor (translation is slow in postmitotic cells such as neurons, and methods to control transcription like sugar promoters or TET promoters either change the metabolic landscape of the cell or are toxic to mitochondria, both represent huge confounding variables). Additionally, TEV is a ‘sloppy’ protease as it cleaves a number of endogenous proteins; this is far from an ideal situation and is why the TEV protease is not commonly used in cells.

What we have done with knockoff is create a method to acutely disrupt a protein of interest at its terminal destination by: (1) demonstrating that the NS3/4 protease can cut a cleavage site that is not immediately adjacent to itself (in fact, it can cleave across large protein domains such as the C2 domains of syt1), (2) screen inhibitors of NS3/4 protease for toxicity and fast dissociation rates, (3) modify the substrate site so that low doses of a fast dissociating inhibitor completely protect the modified protein (100% on or off), and (4) provide evidence that the modified protein is protected from cleavage for weeks at a time (no leak from our system, unlike other systems).